# Venom Variation of Neonate and Adult Chinese Cobras in Captivity concerning Their Foraging Strategies

**DOI:** 10.3390/toxins14090598

**Published:** 2022-08-29

**Authors:** Xuekui Nie, Qianzi Chen, Chen Wang, Wangxiang Huang, Ren Lai, Qiumin Lu, Qiyi He, Xiaodong Yu

**Affiliations:** 1Animal Toxin Group, Engineering Research Center of Active Substance and Biotechnology, Ministry of Education, College of Life Science, Chongqing Normal University, Chongqing 401331, China; 2Key Laboratory of Animal Models and Human Disease Mechanisms of Chinese Academy of Sciences/Key Laboratory of Bioactive Peptides of Yunnan Province, Kunming Institute of Zoology, Kunming 650223, China

**Keywords:** Captive, Chinese cobra, Ontogeny, Snake venom, Proteomics, Transcriptomics

## Abstract

The venom and transcriptome profile of the captive Chinese cobra (*Naja atra*) is not characterized until now. Here, LC-MS/MS and illumine technology were used to unveil the venom and trascriptome of neonates and adults *N. atra* specimens. In captive Chinese cobra, 98 co-existing transcripts for venom-related proteins was contained. A total of 127 proteins belong to 21 protein families were found in the profile of venom. The main components of snake venom were three finger toxins (3-FTx), snake venom metalloproteinase (SVMP), cysteine-rich secretory protein (CRISP), cobra venom factor (CVF), and phosphodiesterase (PDE). During the ontogenesis of captive Chinese cobra, the rearrangement of snake venom composition occurred and with obscure gender difference. CVF, 3-FTx, PDE, phospholipase A_2_ (PLA_2_) in adults were more abundant than neonates, while SVMP and CRISP in the neonates was richer than the adults. Ontogenetic changes in the proteome of Chinese cobra venom reveals different strategies for handling prey. The levels of different types of toxin families were dramatically altered in the wild and captive specimens. Therefore, we speculate that the captive process could reshape the snake venom composition vigorously. The clear comprehension of the composition of Chinese cobra venom facilitates the understanding of the mechanism of snakebite intoxication and guides the preparation and administration of traditional antivenom and next-generation drugs for snakebite.

## 1. Introduction

On 8 April 2019, the WHO uploaded a report on its website about venomous snake bites, a neglected public health problem in many tropical and subtropical countries [1]. Approximately 5.4 million snake bites occur each year, resulting in 1.8 to 2.7 million envenomations from venomous snake bites, 81,410 to 137,880 deaths, and approximately three times as many amputations and permanent disabilities. Most of these bites occur in Africa, Asia, and Latin America. In Africa, approximately 435,000 to 580,000 snake bites require treatment each year. In Asia, 2 million people are bitten by venomous snakes each year. Women, children, and farmers in poor rural areas of low- and middle-income countries are the most affected. The burden is the heaviest in countries with the weakest health systems and scarce medical resources. Approximately 100–200,000 people were bitten each year in China, 70% of whom are young people, with a 5% mortality rate and 25–30% disability rate [2]. The Chinese cobra (*Naja atra*) is one of the major snake species in the epidemiology of snakebites in China [3]. Chinese cobra snakebites are usually accompanied by local swelling, pain, tissue necrosis, palpitations, arrhythmia, shock, cardiac arrest, severe coagulopathy, drooping eyelids, foaming at the mouth, slurred speech, respiratory failure, and skeletal muscle paralysis [4]. Gender [5,6,7], age [3,8,9,10], diet [11,12,13,14], and geographic distribution [15,16,17], may cause intraspecific variation in snake venom, including variations in the protein content and type of venom fractions. The plasticity of snake venom is a great challenge in the therapeutic process of snake bites.

Snake antivenom is the only treatment that is proven to be affective to treat snakebite envenoming. Antivenoms are made of antibodies that are harvested from the serum of hyperimmune animals, typically horse. Thus, the antivenom contains a large amount of heterologous proteins. After the administration of antivenom, heterologous protein induces life-threatening adverse effects such as serum sickness and shock [18]. Antivenom is not always effective against the various complications associated with snake bites [19]. After a snakebite occurs, it is often impossible to accurately determine how much venom of the snake has expelled into the patient’s body through its fangs, or even the type of snake that bit the patient. In addition, there are cases where not all venom components are targeted by antivenom, especially the low molecular mass protein component, three-finger toxin (3-FTx), which is a major contributor to death in patients with snake envenomation [20]. During the treatment of a snakebite, the administration of the dosage and type of antivenom is extremely critical to reduce the possibility of side effects and to guarantee to completely neutralize the corresponding snake venom. To date, the impact of changes in venom components on the clinical symptoms of snakebites has been largely ignored in the treatment of snakebites [18]. At the same time, the standardized production of antivenom does not fully neutralize the venom component of snake venom, which can lead to increased mortality in snakebite patients [21]. Although the next generation of antivenom (recombinant antivenom) that is currently in development is independent of snake species, the formulation and dosing of antivenom are highly dependent on knowledge of the comprehensive venom profile [22]. Therefore, a clear understanding of the snake venom profile is essential for the effective treatment of snakebites.

In recent years, omics studies become popular among Elapidae, Viperidae, and Hydrophiidae [23]. The traditional strategy of snake venom research was to isolate and purify single snake venom proteins, following sequence determination, three-dimensional structural studies, and functional studies. Therefore, the traditional method is inefficient. Genomic, transcriptomic, and proteomic studies of Indian cobras (*N. naja*) have been completed [24,25]; transcriptomic studies of the venom glands of *N. kaothia* and *N. sumatrana* have been reported [15,26]; according to PubMed search results, most of the proteomic studies on the venom of the cobra genus have been reported. Currently, there is no report omics study on captive Chinese cobra. In this study, the venom and venom glands of captive Chinese cobras were studied using LC-MS/MS and Illumina sequencing technology separately, to investigate whether the captive breeding process could reshape the profile of snake venom under captive progress and ontogenesis.

## 2. Results

### 2.1. SDS-PAGE

The electrophoretic profiles of snake venom that were obtained by SDS-PAGE showed a similar pattern of approximately 10 protein bands for each group of snakes under reducing conditions and non-reducing conditions (Figure 1). However, venom from female and male snakes showed age-related variations in their electrophoretic profiles (differences in the intensity of bands). Regarding SDS-PAGE under reducing conditions, partial bands of the crude venom of adult and neonate snakes were distinctly varied. In neonate captive *N. atra* venom, the protein bands of approximately 26, 54, and 66 kDa densimetric more strongly than adult *N. atra* venom. In addition, four protein bands of about 14.4, 34, 45, and 100 kDa were more abundant in the venom of adult *N. atra* venom. This implies that there were differences in certain components of snake venom. However, there were no sex-related differences that showed up under reducing SDS-PAGE.

### 2.2. Venom Protein Concentration and Captive Snake Growth Status

The process of hatching and breeding of *N. atra* is shown in Figure 2A. The standard curve was obtained from the determination of BSA using a modified Bradford method protein concentration assay kit. According to the standard curve, the proportion of protein in each group of crude snake venom samples was calculated (Figure 2B). Indeed, there were no significant differences in the protein proportion of the different groups. There was no significant difference in the body total length between the diverse gender as well (Figure 2C). The average body weight of female neonate and adult *N. atra* was greater than males at the same age, respectively (Figure 2D).

### 2.3. Venom Proteomics

Accurate knowledge of snake venom proteomics contributes to an understanding of the poisoning mechanism of venom components in snakebites. In this study, a venom proteomics protocol was undertaken that included enzyme digestion, HPLC, LC-MS/MS, and sequence matching (Proteome Discoverer 2.4, PD 2.4, Thermo). In all the sample groups, a total of 127 proteins from 21 protein families were identified by MS analysis (Table 1). The proteins were identified by finding at least one unique peptide per protein (Appendix A). The peptides that were used for sequencing after enzymatic hydrolysis ranged in length from 6 to 31 amino acids, with a predominant distribution of 7–14 amino acids (Appendix A). The most abundant protein component in snake venom was over 60% (Appendix A) and had a molecular mass of 0–14 kDa. Our results showed that captive *N. atra* venom mainly consisted of three-finger toxin (3-FTx, 60.87%, 62.22%, 67.99%, and 68.20% of total venom from the neonate male, neonate female, adult male, and adult female group, respectively); cysteine-rich secretory protein (CRISP, 11.39%, 9.37%, 7.08%, and 6.65%, respectively); snake venom metalloproteinase (SVMP, 10.22%, 10.13%, 4.70%, and 4.93%, respectively); cobra venom factor (CVF, 3.53%, 3.92%, 5.02%, and 4.78%, respectively); and phosphodiesterase (PDE, 3.49%, 3.30%, 4.35%, and 4.51%, respectively) (Table 2). The number of proteins that were jointly identified in the four groups was 119. Neonate female venom specimens had higher relative levels of the three finger toxins (neurotoxins and cytotoxins) snake venom proteins and lower relative levels of CVFs and PLA_2_ compared to adult female venom specimens. The venom of the neonate males contained higher relative levels of SVMPs, CTXs, and neurotoxins than did the adult males. There were gender differences in the venom composition of snakes. Neonate male snake venom had higher relative levels of CTX, L-amino acid oxidase (LAAO), muscarinic tox-in-like protein (MTLP), kunitz-type serine protease inhibitor (KUN), but (SNT, CTX) were relatively low. The SVMP content was higher in the female adult specimens (Figure 3).

### 2.4. Protein Sequence Alignment

3-FTx are the main components that make up the snake venom proteins and are responsible for snakebite lethality [27]. According to our results, 3-FTx was most abundant component in *N. atra* snake venom and can be classified into CTX and neurotoxin based on their biological targets of action. There are two subfamilies, CTX and neurotoxin, that were high in content and had massive members in each subfamily. Our results revealed that 3-FTx contains 8–10 cysteine residues for disulfide bond formation and further leads to the formation of the large family of 3-FTx. 3-FTx signal peptides were extremely conserved (Figure 4). The neurotoxin long neurotoxin (LNX) had five pairs of disulfide bonds (Figure 4A), weak neurotoxin (WNX) had five pairs of disulfide bonds (Figure 4B), short neurotoxin (SNX) had four pairs of disulfide bonds (Figure 4C), and CTX had four pairs of disulfide bonds (Figure 3D). LNX binds with high affinity to muscular (alpha-1/CHRNA1) and neuronal (alpha-7/CHRNA7) nicotinic acetylcholine receptor (nAChR) and inhibits acetylcholine from binding to the receptor, thereby impairing neuromuscular and neuronal transmission. P01391was well studied and there were four forms of existence.

In Figure 4C (SNX), the muscarinic acetylcholine receptor antagonists consisted of P18328, P82462, and P82463. The mechanism of toxin-receptor interaction was comprised of at least two steps. The first step is fast with no competition between the toxin and the antagonist [28]. The second step is slow with the formation of a more stable toxin-receptor complex and the inhibition of the antagonist binding. Other SNX bound to muscle nicotinic acetylcholine receptor (nAChR) and inhibit acetylcholine from binding to the receptor, thereby impairing neuromuscular transmission.

In this study, more than 60% of the three-finger toxins were CTX, and they usually induced apoptosis of multiple classes of cells. The main CTX shows cytolytic activity on many different cells by forming a pore in the lipid membranes. In vivo, they increase heart rate or kill the animal by cardiac arrest. In addition, it binds to heparin with high affinity, interacts with Kv channel-interacting protein 1 (KCNIP1) in a calcium-independent manner, and binds to integrin alpha-V/beta-3 (ITGAV/ITGB3) with moderate affinity [29]. However, there were some special CTX. Q53B46 Acts as a beta-blocker by binding to beta-1 and beta-2 adrenergic receptors (ADRB1 and ADRB2). It dose-dependently decreases the heart rate (bradycardia), whereas conventional cardiotoxins increase it [30]. P60305 shows cytolytic activity (apoptosis is induced in C2C12 cells). Basic protein (P01442) that binds to the cell membrane and depolarizes cardiomyocytes (may interact with sulfatides in the cell membrane which induces pore formation and cell internalization) and it also may target the mitochondrial membrane and induce mitochondrial swelling and fragmentation [31]. P07525 functions as same as P01442 and also inhibits protein kinases C [32]. 3-FTx is abundant in Chinese cobras with structural and functional diversity.

Snake venom metalloproteinases (SVMP) are part of the cobra venom. This class of proteins consists of a combination of structural domains, forming assemblies of different structural domains, leading to the formation of structurally and functionally diverse SVMP families. SVMP possesses substantial biological activity. It hydrolyzes human fibrin (pro), acts as a prothrombin activator (activates blood coagulation factor X) [33], inhibits platelet aggregation [34], promotes the inflammatory response of the body [35], inhibits the activity of serine protease inhibitors in the blood [36], and hydrolyzes basement membrane components (laminin, nestin, Type IV collagen) leading to bleeding. SVMP also has pro-inflammatory effects [18]. Cofactors of the SVMP family members (Ca^2+^, Zn^2+^) and glycosylation modifications of proteins (metalloproteases, disintegrin, cysteine-rich structural domains on peptide chains asparagine residues that are linked to glycosyl groups). Glycosylation sites are present on the structural domains of the basal genus protease, disintegrin, and cysteine-rich, but not all three structural domains of the same protein have glycosylation modifications (Table 3). The results show (Figure 5) that the described SVMPs are all P-III metalloproteases, except for P82942 and D6PXE8, which only have 14 pairs of disulfide bonds within the molecule, and the remaining SVMPs, which form 17 pairs of disulfide bonds within the molecule. The active center of all the SVMPs is the amino acid glutamate residue at position 343, only Q10749 at position 343 is a glutamine residue, with opposite acidity.

The PLA_2_ family of snake venom proteins are widely distributed in the Elapidae, Viperidae, and Colubridae. PLA_2_, active forms include dimers and monomers, has a highly conserved Ca^2+^ binding loop (XCGXGG), and active center (DXCCXXHD) in both major structures, corresponding to amino acid numbers 55–60 and 69–76, respectively, in Figure 6. All PLA_2_s contain seven pairs of disulfide bonds according to the sequence comparison in Figure 6. P00598 inhibits G protein-coupled muscarinic acetylcholine receptors and A4FS04 effectively inhibits A-type K^+^ currents (Kv/KCN) in acutely dissociated rat dorsal root ganglion (DRG) neurons. This inhibitory effect is independent of its enzymatic activity. The anticoagulant effect of Q6T179, Q9DF33, and P00598 is through interference with the coagulation cascade. All are acidic PLA_2_s except for P60043 and P00599, which are basic PLA_2_s. The active sites are located at positions 75 (Histidine) and 121 (Aspartic acid).

### 2.5. Venom Gland Transcriptome

Revealing the transcriptome profile of snake venom glands is essential to decipher the functional role of snake venom, which was performed by next-generation sequencing. In our results, from neonate male, neonate female, adult male, and adult female groups, clean reads and unique gene (FPKM > 1) were identified, 43,700,115 and 11,153, 42,159,554 and 10,872, 42,965,160 and 10,456, 45,060,349, and 10,600, respectively. In the neonate male, neonate female, adult male, adult female groups, 70% of the paired regions were exons, the rest of the paired regions were intron and intergenic. However, we found 98 shared snake venom unique genes, among these unique genes were contained 10 new transcripts and 88 unique transcripts (FPKM > 1) (Table 4). Our results showed that the *N. atra* venom gland transcriptome mainly consists of 3-FTx (85.31, 88.22, 87.82, and 80.40% of the total transcriptome from the neonate male, neonate female, adult male, and adult female group, respectively); PLA_2_ (1.31, 1.25, 3.41, and 1.83%, respectively); natriuretic peptide (NP, 4.74, 3.81, 2.56, and 6.93%, respectively); snake toxin and toxin-like protein (STLK, 1.67%, 2.18%, 1.85%, and 3.06%, respectively); and glutathione peroxidase (GPX, 2.35, 1.52, 1.39, and 2.59%, respectively) (Table 5). Based on the results of bioinformatics analysis, the total percentage of low-abundance expression of the transcripts of venom components in the four groups was less than 4% (Table 5).

### 2.6. Cooperative Proteomic and Transcriptomic Analysis

Among all the captive *N. atra* (neonate male, neonate female, adult male, and adult female groups), we found that high correlations only for the transcriptomic and proteomic statistical analysis of 3-FTx among dominant snake venom families, but significantly different levels for other snake venom families.

3-FTx, a non-enzymatic snake venom protein, is present in the majority of Elapidae snake venoms and contains a variety of biological activities. [37]. Based on the length of the peptide chain and the biological target of the 3-FTx, 3-FTx can be classified into five subclasses, cytotoxins (CTX, 40.00% neonate male group, 40.22% neonate female group, 51.77% adult male group, and 48.63% adult female group, respectively) including 26 proteins; weak neurotoxins (WNX, 5.08, 6.18, 5.09, and 5.79%, respectively) including 7 proteins; long-neurotoxins (LNX, 0.10, 0.09, 0.13, and 0.14%, respectively) including 3 proteins; and short-neurotoxins (SNX, 15.67, 15.73, 11.00, and 13.65%, respectively) including 20 proteins (Table 1 and Table 2). The above results verified the 3-FTx plays an important role in the envenomation of *N. atra* that is responsible for the death of victims. Our results showed that the correlation between the changes in the proteome and transcriptome of cultured Chinese cobra snake venom was not significant (*p* > 0.05), except for neonate females versus adult females (Figure 7). Indeed, many previous studies lack concordance between transcriptome and proteome, including *Boiga irregularis* [38], *N. kaouthia* [39], *Crotalus simus* [40].

A total of 37 proteins of venom were matched to one or more transcript sequences using BLAST, while the vast majority of other proteins did not match the corresponding transcripts (Table 6).

### 2.7. Toxicological and Enzymatic Activity of N. atra Venom

To characterize the relationship of the dose-effects between the snake venom protein families and the corresponding functional assays, we performed toxicological and enzymatic activities of *N. atra* crude venom. 3-FTx is thought to act on the neuromuscular junction, acetylcholine receptors, and muscarinic receptors to achieve a blocking effect on nerve signaling transmission. The activity of SVMP, LAAO, and 5′-nucleotidase (5′-NT) of neonate *N. atra* was higher than the adult and without a gender difference (Figure 8A,B,D). There was a higher toxicity in the adult specimens. There were age differences (adult > neonate) but no sex differences in PLA2 activity in *N. atra* crude toxin (Figure 8C); there were neither sex nor age differences in AChE activity (Figure 8E). The LD_50_ of crude venom of the neonate male, neonate female, adult male, and adult female groups were 0.73, 0.74, 0.82, and 0.82 μg/g (intraperitoneal injection), respectively (Figure 8F). All of the tested enzymes indicated a good correlation (*p* < 0.05) between enzyme activity and enzyme content in each group (Figure 9A–E).

## 3. Discussion

### 3.1. Ontogenetic Variation Predominates in the Snake Venom of Captive Species

Our results indicated that there were age differences in the venom composition of captive Chinese cobras. The relative contents of SVMP and CRISP were higher in the neonates; the relative contents of 3-FTX, CVF, and PLA2 were higher in the adults. This result is generally consistent with the variation of venom composition in neonate and adult specimens of wild cobras in Guangxi Province [3]. The results of SDS-PAGE electrophoresis of the crude venom of the snakes (Figure 1) indicated age differences in the venom content. Furthermore, the results of enzyme activity and semi-lethal concentration assay further confirmed this (Figure 8 and Figure 9). It is noteworthy that the changes in the relative content of the transcripts and proteins were consistent only for PLA_2_. In addition, the snake venom components that showed gender differences included AChE (male specimens > female specimens), NGF (male specimens < female specimens), and CRISP (male specimens > female specimens). In our study, only AChE was more abundant in male specimens than that in the female specimens, and both adult specimens and neonate specimens. Based on the transcript of venom toxin distribution of toxin genes on chromosomes, no toxin-related genes were found on the W and Z sex chromosomes. Therefore, the genes for these sexually distinct snake venom components are not distributed on the sex chromosomes and the cause of the appearance of sex differences is not companionship. Other non-genetic factors may be suspected to cause the sex differences in the toxin.

Previous studies have shown that macroscopic factors contributing to changes in snake venom composition during species ontogeny include prey species (ectothermic prey or endothermic prey) [41], surface area to volume ratio (same prey species during ontogeny) [8], ontogenetic shifts in dietary habits, competition, predation pressure [3], and microscopic factors, such as miRNA [42,43]. In captivity, the environmental factors for neonate and adult snakes were the same except for the food, which was thought to be the same, and the adult and neonate snakes were fed block chicks and frogs (*Pelophylax nigromaculatus*), respectively. Amphibian reptiles are variable temperature animals, while chicks are constant temperature animals. The prey for wild Chinese cobra neonates is mainly amphibious reptiles and small-sized fish, while the prey for adult snakes are mainly birds and mice. Therefore, the difference in venom fractions during ontogenesis may be due to the type of food (ectothermic prey or endothermic prey).

According to the results in Figure 6, the relative changes of snake venom proteins during the growth and development of individuals in captive-bred Chinese cobras were poorly correlated with the transcripts of the corresponding snake venom proteins. There are several possible reasons for this result. First, the limited number of samples that were taken and the variability of the sampled individuals. Second, the venom glands were sampled at an inappropriate time. Third, inaccuracies in the proteomic or transcriptional sequencing process. Fourth, improper handling and storage of proteomic and transcriptomic samples resulted in the degradation or contamination of the samples.

We confirmed at the transcriptomic and proteomic levels that all snake venom metalloproteases in Chinese cobra venom were Class III SVMPs. Class III SVMPs exhibit complex biological activities due to the diversity of molecular structures (metalloproteinase, deintegrin, cysteine-rich domain). Another source of Class I and Class II SVMPs in snake venom, in addition to the corresponding gene expression, is the self-hydrolysis process of Class III [44]. However, the disappearance of metalloproteinase self-hydrolysis may be due to its own glycosylation modifications (Table 3). The snake venom proteases are associated with prey digestion (cytotoxins and hydrolases) [45]. For example, SVMP takes responsibility for the local and systemic hemorrhage by hydrolyzing basement membranes, extracellular matrix, promoting apoptosis, inhibiting platelet aggregation, hydrolyzing collagen, fibrinogen, and also inducing leukocyte rolling [44,46]. The destruction of blood vessels may promote the penetration and diffusion of other toxins and accelerate the digestion process of prey. It is assumed that the neonate snake’s strategy to ensure survival is rapid growth. What was puzzling was that females weighed significantly more than the males at the same growth period, but there were no gender differences in the protein hydrolase relative content levels. Moreover, there was no significant difference in the body lengths between the sexes. Therefore, the levels of cytotoxic and hydrolytic enzymes might not be directly related to the body length and weight.

The relative content of CRISP protein in snake venom of captive Chinese cobras is rich. CRISP is widely distributed in snake venom and acts on ion channels, the inflammatory response, and smooth muscle contraction [47]. Snake venom CRISPs with a molecular weight between 20 and 30 kDa are non-enzymatic proteins, containing 16 highly conserved cysteine residues forming eight disulfide bonds, four of these residues form a cysteine-rich domain (CRD region) at the C-terminus. Despite the richness of the biological targets of action of CRISP, its specific function remains unclear. Since CRISP can promote smooth muscle contraction, it is presumed that CRISP may act on the circulatory system, to raise blood pressure and promote increased hemorrhage. The silica molecular docking study of cobra CRISP in complex with TLR4-MD2 receptor shows that CRISP is involved in its cysteine-rich structural domain (CRD) interacting with the complex [48]. The rich diversity of CRISP in intra- and inter-species snake venoms suggests functional diversity. Dissecting these rich structural and functional CRISPs promotes an understanding of the mechanism of action of snakebites and exploits resources for drug development. Surprisingly, the relative levels of CRISP in the venom of captive-bred individuals increased dramatically compared to the wild individuals, but the exact reason for this is not clear.

CVF leads to a greater understanding of complement C3. CVF is a spherical glycoprotein, a homologous molecule that is formed in evolution with complement C3, similar in structure to C3a, and reversibly bound to factor B in the presence of Mg^2+^. The complex CVF-B is then hydrolyzed by factor D in serum into two fragments, CVF-Bb and Ba. CVF-Bb can be present in the blood for 6–7 h and the small fragment Ba is released. The product CVF-Bb is a complement-activated alternative pathway C3/C5 convertase with serine protease activity that hydrolyzes the peptide bond at position 77 of the alpha chain of C3 (Arg-Ser) and the peptide bond at position 74 of the alpha chain of C5 (Arg-Leu) to form C3b and C5b, releasing the allergenic toxins C3a and C5a. C5b contributes to the formation of the membrane attack complex, leading to cell membrane perforation and cell lysis. It also caused the depletion of C3 and C5–C9 components, and C3 was largely depleted in the blood of rats after 2 h of intraperitoneal injection of CVF (1 mg/kg) [49]. Finally, CVF weakens complement resistance to other snake venom proteins and assists other toxins in their toxicity.

Snake venom PDEs are among the least studied snake venom enzymes. During envenomation, they exhibit various pathological effects, ranging from the induction of hypotension, to the inhibition of platelet aggregation, edema, and paralysis [50]. In the present study, only one phosphodiesterase was obtained and was not an abundant component. In addition, no PDE transcripts were found in the venom gland transcriptome of wild *N. atra* that were collected from Zhejiang Province [51].

PLA2 is distributed among various families of snakes. Generally, PLA_2_ is exceptionally plentiful in biological activities, including neurotoxicity, hemorrhagic toxicity, myotoxicity, cardiotoxicity, and the induction or inhibition of platelet aggregation. In our study, the proportion of acid PLA_2_ content was significantly higher than that of alkaline PLA_2_. The active catalytic center of all PLA_2_ matched was His48-Asp49-Tyr52, the motif underlying the high activity of PLA_2_. There are two major amino acids, Lys115 and Arg117, that are thought to underlie the neurotoxicity of these enzymes and indicated that 3 out of 11 PLA_2_ has neurotoxicity among all the groups. The conserved structures of these enzymes are essentially the same in different snake species they have the same evolutionary origin. They both have 14 cysteine residues and form seven disulfide bonds. According to the results of previous studies, *N. atra* expressed up to 12.2% PLA_2_ in their venom in the wild (n = 13, all adults), while the expression of PLA_2_ in the venom of the artificially hatched individuals in this study was less than 1% (among all groups) [52].

### 3.2. Neonate and Adult Snakes have Different Strategies for Handling Prey

The biological functions of snake venom include predation, defense, and digestion [53]. In the process of prey predation, snake venom neurotoxins play a key role in immobilizing and then killing the prey, increasing the efficiency of prey predation, which likewise is the synergistic effect of 3-FTx and fewer abundance components in the venom [54,55]. Different components of snake venom play different biological functions, among which enzymes have the role of helping to digest prey, and 3-FTx is mainly used to kill prey. In the present study, the content of 3FTx was lower in the neonates than in the adults, while the content of enzymes was higher in the neonates than in the adults. The predation strategy of the cobra family is to violently control the movement of the prey after completing the injection of venom into the prey until the toxicity kicks in and the prey is incapacitated. In addition, the prey of neonate Chinese cobras is mainly small-sized amphibians, while the prey of adults is mainly rodents and reptiles [56]. Therefore, the venom pattern of neonate and adult Chinese cobras differs, with neonates having high enzymatic activity and low toxicity and adults having low enzymatic activity and high toxicity. High doses of neurotoxin are incompatible with high protein hydrolytic activity [57]. The LD_50_ of venom from adult Chinese cobras was significantly lower than that of juveniles when measured in mice. In conclusion, the predation strategies of adult and neonate captive *N. atra* were different.

### 3.3. The Hatchery Breeding Process Rearranges Snake Venom

Snake venom composition is regulated by intrinsic factors and also environmental factors. Differences in the venom fractions between captive-bred and wild individuals have been reported for both cobras and vipers [58,59,60,61,62]. However, a comparison of the proteome of venom from long-term captive and recently wild-caught eastern brown snakes (*Pseudonaja textilis*) showed that the venom was not altered by captivity. In this study, the humidity (70%) and temperature (28 °C) were constant during captive breeding, food was the same, and the snakes went without hibernation. The habitat of wild Chinese cobras is more complex and variable. The protein number of wild Chinese cobra venom that was determined by Shotgun-LC-MS/MS, 1DE-LC-MS/MS, GF-LC-MS/MS, and GF-2DE-MALDI-TOF-MS were 78, 65, 77, and 16, respectively [63]. However, this study lacked the description of protein sequence and protein function annotation. In addition, the venomic profile of Chinese cobra that separately inhabited eastern and western Taiwan were described and the results indicated that dominating component contents both were the same and but the proportion of protein families were not equal. The most components of the eastern inhabitants are 3-FTx, phospholipase A_2_ (PLA_2_), and cysteine-rich secretory protein (CRISP) (76.35%, 16.82%, and 2.41%, respectively) and the western inhabitants were 3-FTx, PLA_2_, and CRISP (79.96%, 13.95%, and 2.16%, respectively) [16]. The main components of adult *N. atra* (Zhoushan, Zhejiang province, China) venom was 3-FTx, PLA_2_, CRISP, SVMP, and nerve growth factor (NGF) (84.3, 12.2, 1.8, 1.6, and 0.1%, respectively) [52]. A similar proteome profile was discovered in *N. kouthia* that was caught in China [39]. The transcriptome profile of the Chinese Cobra showed that the 3-FTx was 95.80%, followed by PLA_2_ 1.20%, CRISP 0.70%, C-type lectins 0.60%, and vespryn 0.50% [51]. In our study, the consequence of the venomic (Table 2) and transcriptomes (Table 5) of captive Chinese cobras were distinct from the wild ones about the previous description of snake venom composition. The main components of wild Chinese cobra venom are 3-FTx, PLA_2_, and CRISP. Although 3-FTx was still the main component of captive Chinese cobra venom, its content was significantly lower than that of wild specimens; however, PLA_2_ was not the main component, and its content was nearly ten times lower. Although CRISP was the main component of both cultured and wild specimens, the content of CRISP in captive specimens increased by more than two times compared with that of the wild specimens. Therefore, the captive breeding process with strictly controlled conditions may alter snake venom partial composition.

### 3.4. Asynchronous Snake Venom Synthesis Process

The process of snake venom secretion involves the secretion of a complex mixture of peptides and proteins by secretory columnar cells of the glandular epithelium and storage in the lumen of the venom gland. The process of venom release involves muscle contraction of the venom gland to expel the venom through the fangs. However, there are different views on the origin of the venomous gland apparatus. The venom gland of the king cobra is rich in vertebrate pancreas-specific miRNA (miR-375), confirming that its venom gland originates from the pancreas [64]. In contrast, Kochva proved that the snake venom gland has been adapted from the salivary gland during vertebrate evolution [65]. Heterogeneity of venom gland cells in snakes was observed in king cobra (*Ophiophagus annah*) and horned coral snake (*Aspidelaps lubricus cowlesi*) [64,66]. In addition, the secretory processes of both the snake venom gland and the pancreas are under hormonal and neurological control [67,68,69]. The snake venom that is stored in the lumen is secreted by the venom gland secretory cells after a process of transcription, translation, and modification. The transcript directs the synthesis and modification of the toxin, and the transcript is hydrolyzed by nucleic acid endonucleases once the toxin synthesis process is completed [70]. By comparing the protein sequences of snake venom with the transcript sequences, we found that some of the transcripts of the toxins were absent from the secretory cells of the venom glands and only the corresponding proteins remained, or the transcripts were present but the proteins were missing (Table 1, Table 4 and Table 6). These results suggest that there was a sequential order in the synthesis and secretion of toxins. It is hypothesized that the response times of different toxic gland secreting cells to toxic gland milking stimuli differ.

### 3.5. Widespread Distribution of Toxin Gene on Chromosomes

Snake venom, which contains more than 20 well recognized protein families (also including many non-toxic proteins), is a result of different protein families being recruited to the venom gland at different times in the evolution of the snake. The generation and diversity of snake venoms are attributed to gene recruitment, doubling, and neogenization [71,72]. The vast majority of snake venom genes have been shown to originate from gene recruitment during venom evolution, and toxin recruitment events have been found to have occurred at least 24 times [73,74].In addition, the process of snake venom recruitment seems to be uninterrupted, with new members of the snake venom metalloproteinase family recently reported in rear-fanged snake venom [72].

Genomic data and karyotype maps of Indian cobras were used as a reference for our study. We found that the majority of Chinese cobra snake venom genes were distributed on macro-chromosomes (2n = 38) 1–4, 6, and 7, micro-chromosomes (11 pairs) 1–9 and 11, and without toxin genes were found on the sex chromosomes (Table 4). The gene distribution of the 3-FTx, DPP-IV, PDE, VPA, NP, LAAO, C3, and AP family is located on macro-chromosome 3, 1, 1, 6, 4, 2, 2, and micro-chromosome 6, respectively. Since all 3-FTx genes are localized on the same chromosome, it is speculated that this gene family is amplified by tandem duplication [46]. The gene distribution of the other protein families is scattered across multiple chromosomes, for instance, CTL (macro-chromosome 1, 2, and 7, micro-chromosome 7), VEGF (macro-chromosome 1, 2, 4, micro-chromosome 5, and 9), SVMP (macro-chromosome 4, micro-chromosome 1, 6, and 8), and PLA_2_ (macro-chromosome 1 and 4, micro-chromosome 11). The snake venom gene family expands diversely. These data on the genetic distribution of snake venom protein genes on chromosomes provide the basis for the theoretical analysis of the evolutionary study of toxins and evidence that toxin recruitment and toxin family expansion have occurred over a long period and through a large number of evolutionary events.

To date, transcriptomics and proteomics technology have been widely used in snake venom research. In the past decades, a large number of toxic or non-toxic venom proteins have been sequenced and characterized for physicochemical properties and function. With the popularity of toxin proteomics, a large number of proteins regarding snake venom components have been identified at a very fast rate [75]. In snake venom and venom gland studies, the combination of venomics and venom gland transcriptomics is increasingly being applied jointly to find evidence of intra- and inter-specific individual differences in snake venom secretion species, and the combination of these two omics approaches helps to explain the differences that already exist more scientifically and rigorously. All snakebite treatments rely on accurate snake venom protein profiles, including antivenom, next-generation treatments (including monoclonal antibodies and antibody fragments, nanobodies, small molecule inhibitors, aptamers, and peptides, metal ion chelators, and antivenoms that are manufactured using synthetic immunogens) [76]. In the present study, a combined transcriptomic and proteomic approach was adopted to study the corresponding changes of captive snakes in ontogenesis. Based on the results of our study, we found that 3-FTx relative content varied in a discrete and uncorrelated manner between 3-FTx transcriptional and proteomic levels among different ages and gender specimens. Moreover, the same trend of variation was not synchronous in the relative content of 3-FTx subclasses. The same was true for several other major components of snake venom, the CRISP, CVF, SVMP, PDE, and NGF. The relative content of the venom protein family varied considerably among the captive *N. atra* individuals of different ages, and secondly, the venom composition between the captive and wild individuals also differed significantly.

## 4. Conclusions

In the present research, the transcriptome, proteome, and functional assays of captive *N. atra* venom was depicted. Under the captive conditions, females grew faster than males during ontogenesis. Ontogenesis has an overwhelming effect on the venom protein profile of captive *N. atra*, while there is an obscure gender difference. The main components of captive *N. atra* venom was very different from the wild specimens, although they were obtained from the same geographical distribution area, Guangxi province, China. Presumably, the loss of transcripts of some toxin proteins indicates that the expression of different snake venom families was asynchronous under the same toxin extraction stimulation conditions. An analysis of the proteomic and transcriptomic results revealed significant effects of ontogenesis and captive breeding on the relative content of the snake venom protein, demonstrating the plasticity of snake venom expression and secretion. The protein profile of snake venom provides the foundation for the formulation and administration of traditional antivenom and next-generation antivenom.

## 5. Materials and Methods

### 5.1. Animals Keeping and Feeding

All the experimental procedures involving animals were carried out under the Chinese Animal Welfare Act and our protocol (20090302 on 10 January 2021) was approved by Chongqing Municipal Public Health Bureau. The healthy female Kunming mice (20 ± 2 g of body weight) were obtained from the Laboratory Animal Center of the Army Medical University. They were housed in temperature-controlled rooms and received water and food ad libitum until use.

Snake eggs were obtained from Guangxi province, China. All the eggs were incubated and all snakes were captive in our laboratory without hibernation, Chongqing, China. Newborn snakes were fed with a starter diet (gutted and chopped frog) for three months and a gradual increase in the feeding of dehaired, frozen chicks (gutted and chopped). The feeding frequency was once every two days. The constant ambient temperature was 28 °C, the humidity was 70%, and a supply of sterilized water. An average of 50 snakes in a four-square meter enclosure, which were enclosed in nylon netting.

### 5.2. Snake Venom and Gland

A total of 10 snakes in each group (neonate male, neonate female, adult male, and adult female group), were randomly selected to measure the body total length and body weight. The venom milking process was that snake bites on parafilm-wrapped jars along with gently massaging the venomous glands or aspirate snake venom with a pipette aimed at the fangs. A total of five snakes were randomly selected from each group (neonate male, neonate female, adult male, and adult female group) to collect venom and venom glands. The fresh venom was centrifuged to remove impurities for 15 min at 10,000 *g* 4 °C, lyophilized, marked, and stored at −80 °C until use. During the lyophilization process, the lyophilizer that was used was a SCIENTZ-18N (SCIENTZ, Ningbo, China) with the condition of −70 °C sample temperature and −45 °C cold trap temperature. The venom gland was dissected four days after the venom was milked and stored in RNA later solution at −80 °C until use.

### 5.3. Library Preparation for Transcriptome Sequencing

A total amount of 1 µg RNA per sample was used as the input material for the RNA sample preparations. Sequencing libraries were generated using NEBNext^®^ UltraTM RNA Library Prep Kit for Illumina^®^ (NEB, San Diego, CA, USA) following the manufacturer’s recommendations and index codes were added to attribute sequences to each sample. Briefly, mRNA was purified from the total RNA using poly-T oligo-attached magnetic beads. Fragmentation was carried out using divalent cations under elevated temperature in NEBNext First Strand Synthesis Reaction Buffer (5×). First-strand cDNA was synthesized using random hexamer primer and M-MuLV Reverse Transcriptase. Second strand cDNA synthesis was subsequently performed using DNA Polymerase I and RNase H. The remaining overhangs were converted into blunt ends via exonuclease/polymerase activities. After adenylation of 3′ ends of DNA fragments, NEBNext Adaptor with hairpin loop structure was ligated to prepare for hybridization. To select cDNA fragments of preferentially 250–300 bp in length, the library fragments were purified with the AMPure XP system (Beckman Coulter, Beverly, CA, USA). Then, 3 µL USER Enzyme (NEB, San Diego, CA, USA) was used with size-selected, adaptor-ligated cDNA at 37 °C for 15 min followed by 5 min at 95 °C before PCR. Then PCR was performed with Phusion High -Fidelity DNA polymerase, Universal PCR primers, and Index (X) Primer. At last, the PCR products were purified by AMPure XP system(Beckman Coulter, Beverly, USA) and the library quality was assessed on the Agilent Bioanalyzer 2100 system (Agilent, Santa Clara, CA, USA).

### 5.4. cDNA Sequencing and Data Analysis

The clustering of the index-coded samples was performed on a cBot Cluster Generation System using TruSeq PE Cluster Kit v3-cBot-HS (Illumia, San Diego, CA, USA) according to the manufacturer’s instructions. After cluster generation, the library preparations were sequenced on an Illumina Novaseq platform and 150 bp paired-end reads were generated. At the same time, the Q20, Q30, and GC content of the clean data were calculated. All the downstream analyses were based on clean data with high quality. Reference genome and gene model annotation files were downloaded from the genome website directly. Index of the reference genome was built using Hisat2 v2.0.5 and paired-end clean reads were aligned to the reference genome (GenBank GCA_009733165.1) using Hisat2 v2.0.5. StringTie uses a novel network flow algorithm as well as an optional de novo assembly step to assemble and quantitate the full-length transcripts representing multiple splice variants for each gene locus. FeatureCounts v1.5.0-p3 was used to count the reads numbers that are mapped to each gene. The FPKM of each gene was calculated based on the length of the gene and the read counts that were mapped to this gene. Differential expression analysis of two conditions/groups (two biological replicates per condition) was performed using the DESeq2 R package (1.16.1, Bioconductor, Seattle, DC, USA, 2005). The resulting *p*-values were adjusted using Benjamini and Hochberg’s approach for controlling the false discovery rate. Genes with an adjusted *p* < 0.05 found by DESeq2 were assigned as differentially expressed.

### 5.5. TMT Labeling of Peptides

Each equivalent snake protein sample was taken and the volume was made up to 100 µL with DB dissolution buffer (8 M Urea, 100 mM TEAB, pH 8.5). Trypsin (Promega/V5280, Madison, WI, USA) and 100 mM TEAB buffer (Sigma/T7408-500ML, St. Louis, MO, USA) were added, the sample was mixed and digested at 37 °C for 4 h. Then, trypsin and CaCl_2_ were added and the sample was digested overnight. Formic acid was mixed with the digested sample, adjusted pH under 3, and centrifuged at 12,000 *g* for 5 min at room temperature. The supernatant was slowly loaded to the C18 desalting column (Waters BEH C18, 4.6 × 250 mm, 5 μm, Milford, MA, USA), washed with washing buffer, including 0.1% formic acid (Thermo Fisher Scientific/A117-50, Waltham, MA, USA), 3% acetonitrile (Thermo Fisher Chemical/A955-4, Waltham, MA, USA), 3 times, then eluted by some elution buffer (0.1% formic acid, 70% acetonitrile). The eluents of each sample were collected and lyophilized. A total of 100 µL of 0.1 M TEAB buffer was added to reconstitute, and 41 µL of acetonitrile-dissolved TMT labeling reagent was added, the sample was mixed with shaking for 2 h at room temperature. Then, the reaction was stopped by adding 8% ammonia. All the labeling samples were mixed with equal volume, desalted, and lyophilized.

### 5.6. Separation of Fractions

Mobile phase A (2% acetonitrile, adjusted pH to 10.0 using ammonium hydroxide) and B (98% acetonitrile) were used to develop a gradient elution. The lyophilized powder was dissolved in solution A and centrifuged at 12,000 *g* for 10 min at room temperature. The sample was fractionated using a C18 column (Waters BEH C18, 4.6 × 250 mm, 5 µm, Milford, MA, USA) on a Rigol L3000 HPLC system (Rigol, Shanghai, China), the column oven was set as 45 ℃. The detail of the elution gradient is shown in Appendix A. The eluates were monitored at UV 214 nm, collected for a tube per minute, and combined into 10 fractions finally. All the fractions were dried under vacuum, and then, reconstituted in 0.1% (*v*/*v*) formic acid in water. During the lyophilization process, the lyophilizer that was used was a SCIENTZ-18N (SCIENTZ, Ningbo, China) with a sample temperature of −70 °C and a cold trap temperature of −45 °C.

### 5.7. LC-MS/MS Analysis

For transition library construction, shotgun proteomics analyses were performed using an EASY-nLCTM 1200 UHPLC system (Thermo Fisher, Waltham, MA, USA) coupled with a Q ExactiveTM HF-X mass spectrometer (Thermo Fisher, Waltham, MA, USA) operating in the data-dependent acquisition (DDA) mode. A total of 1 µg sample was injected into a homemade C18 Nano-Trap column (4.5 cm × 75 µm, 3 µm). The peptides were separated in a homemade analytical column (15 cm × 150 µm, 1.9 µm), using a linear gradient elution as listed in Appendix A. The separated peptides were analyzed by Q ExactiveTM HF-X mass spectrometer (Thermo Fisher, Waltham, MA, USA), with ion source of Nanospray Flex™ (ESI), spray voltage of 2.3 kV, and ion transport capillary temperature of 320 °C. Full scan ranges from *m*/*z* 350 to 1500 with the resolution of 60,000 (at *m*/*z* 200), an automatic gain control (AGC) target value was 3 × 10^6^, and a maximum ion injection time was 20 ms. The top 40 precursors of the highest abundant in the full scan were selected and fragmented by higher-energy collisional dissociation (HCD) and analyzed in MS/MS, where the resolution was 45,000 (at *m*/*z* 200) for 10 plex, the automatic gain control (AGC) target value was 5 × 104 the maximum ion injection time was 86 ms, the normalized collision energy was set as 32%, the intensity threshold was 1.2 × 10^5^, and the dynamic exclusion parameter was 20 s.

### 5.8. The Identification and Quantitation of Protein

The resulting spectra from each run were searched separately against the Uniprot database by Proteome Discoverer 2.4 (Thermo, Waltham, MA, USA). The searched parameters are set as follows: mass tolerance for precursor ion was 10 ppm and mass tolerance for production was 0.02 Da. Carbamidomethyl was specified as fixed modifications, oxidation of methionine (M) and TMT plex were specified as dynamic modification, acetylation, and TMT plex were specified as N-terminal modification in PD 2.4. A maximum of 2 miscleavage sites was allowed. To improve the quality of analysis results, the software PD 2.4 further filtered the retrieval results and peptide spectrum matches (PSMs) with the credibility of more than 99% were identified PSMs. The identified protein contained at least 1 unique peptide. The identified PSMs and protein were retained and performed with FDR no more than 1.0%. The protein quantitation results were statistically analyzed by *t*-test.

### 5.9. Protein and Transcript Alignment

The protein family members sequence alignment was performed using online multiple sequence alignment analysis (MAFFT version 7) with default parameters. The protein sequence and transcriptome sequence alignment analysis were performed using BLASTx, E-value < 1 × 10^−5^, with the other parameters set as default.

### 5.10. Snake Protein Concentration Determination

The protein concentration of each snake venom was obtained using the Pierce ^®^ Bicinchoninic Acid (BCA) Protein Assay (Thermo Scientific, Rockford, IL, USA) following the manufacturer’s instructions and using bovine serum albumin (BSA) as a standard.

### 5.11. SDS-PAGE

A total of 40 μg of 15 μL snake protein sample was loaded to 15% SDS-PAGE gel electrophoresis, wherein the concentrated gel was performed at 80 V for 20 min, and the separation gel was performed at 120 V for 90 min. Electrophoresis instrument (BIO-RAD/PowerPac Basic, Hercules, CA, USA) and electrophoresis tank (Bei-jing JUNYI DONGFANG/JY-SCZ2+, Beijing, China) were used in this research. The gel was stained by Coomassie brilliant blue R-250 (Beyotime Biotechnology, Jiangsu, China) and decolored with elution solution until the bands were visualized clearly by a gel reader (Beijing Liuyi/WD-9406, Beijing, China).

### 5.12. Median Lethal Dose (LD_50_) Determination

Groups of 5 healthy female Kunming mice (20 ± 2 g) from the Laboratory Animal Center of Army Medical University were intraperitoneally injected with various doses of captive crude *N. atra* venom, which was dissolved in 100 μL sterile saline, while the control test with an equal volume of sterile saline injection only. The deaths were recorded 6 h after injection, and the LD_50_ was calculated using the Spearman–Karber method.

### 5.13. Enzyme Activity Assay

For the SVMP activity assay [77], 0.5 mL of substrate (0.2 M Tris-HCl, pH 8.5, con-taining 1% azo-casein) was added to the snake venom solution (30 µg dissolved in 10 µL of dd water), mixed well, and incubated at 37 °C for 1 h. A total of 75 μL of 10% TCA(Sigma/T6508-100ML, St. Louis, MO, USA) was added and incubated at 37 °C for 30 min to complete the reaction. An equal amount (100 μL) of supernatant was collected, 50 μL of 2 M Na_2_CO_3_ (Sigma/5330050050, St. Louis, MO, USA)was added and mixed, and the absorbance values of the samples were measured at 440 nm (Flash ^®^ SP-Max2300A2, Shanghai, China).

For the PLA2 activity assay [78], the preparation of egg yolk solution was conducted as follows: the egg yolk was mixed in 0.85% NaCI (SCR/10019318-500g, Shanghai, China) solution at a ratio of 1:3, centrifuged at 3000 r/min for 5 min (IKA ^®^ VORTEX, Guangzhou, China), the supernatant was removed, and left to stand at −20 °C. For the plates preparation: add 0.05 M pH 6.5 NaAC (SCR/10018818-500g, Shanghai, China) solution, add 1% agarose (SCR/10000561-250g, Shanghai, China) without shaking and microwave for 90 s. Reduce the temperature of the agarose to 50 °C, add 4% egg yolk solution and 2% 0.01 M CaCl_2_ (SCR/20011160-500g, Shanghai, China), mix the mixture well, and spread it on the plates. The mixture was allowed to cool to room temperature in the plate, and then the plates were punched and 30 μg venom was loaded per well. After incubation at 37 °C for 8 h, the diameter of the transparent circle was measured using vernier calipers.

For the LAAO activity assay [79], a total of 30 μg venom was added to 90 μL of the substrate system, containing 50 mM Tris-HCl (Sigma/T5941-100g, St. Louis, MO, USA), pH 8.0, 5 mM L-leucine (Sigma/L8000-25g, St. Louis, MO, USA), 2 mM O-phenylenediamine (Sigma/P23938-5g, St. Louis, MO, USA), and 0.81 U/mL horseradish peroxidase (Sigma/P8375-5KU, St. Louis, MO, USA), mixed, and incubated at 37 °C. After one hour of incubation, the reaction was stopped by adding 50 μL of 2 M sulfuric acid. The absorbance was recorded at 490 nm (Flash ^®^ SP-Max2300A2, Shanghai, China). The change in the absorbance at 0.3 units/min/μg venom was used to express the activity.

For the 5′-NT activity assay [80], a total of 40 μg venom was added to 90 μL of substrate solution, containing 50 mM Tris-HCl, pH 7.4, containing 10 mM MgCl_2_ (Sigma/M8266-100g, St. Louis, MO, USA), 50 mM NaCl, 10 mM KCl (Sigma/P3911-25g, St. Louis, MO, USA), and 10 mM 5′-AMP (Sigma/01930-5g, St. Louis, MO, USA), incubated at 37 °C for 60 min, then 20 μL of 2.5 M sulfuric acid was added and mixed and stood for 5 min, and 80 μL of chromogenic solution (8% The absorbance values were measured at 660 nm (Flash ^®^ SP-Max2300A2, Shanghai, China). The H2PO4- activity was expressed as nmol /min/mg of venom that was produced, using kH2PO4 (SCR/10017605-250g, Shanghai, China) as the standard.

For the AChE activity assay [81], the substrate solution contained 75 mM acetylcholine iodide and 10 mM DTNB (Sigma/D8130-1g, St. Louis, MO, USA) that was dissolved in 0.1 M PBS (pH 8.0) buffer containing 0.1 M NaCl and 0.02 M MgCl_2_.6H_2_O, respectively. A total of 100 μL of substrate was added to each 96-well plate, 30 μg of snake venom was added immediately, and the reaction was carried out at 37 ℃ for 30 min. The samples were vortexed gently for 15 s (IKA ^®^ VORTEX, Guangzhou, China) and the absorbance values were measured at 412 nm (Flash ^®^ SP-Max2300A2, Shanghai, China).

Each of the enzyme activity enzyme activity test methods were simply modified.

### 5.14. Statistical Analyses

There were three replicates per treatment that were used in all the studies. The means were obtained by taking the average of three measurements for each experiment with the standard error of the means (±SE; standard error) obtained. An analysis of variance (ANOVA) was applied to analyze the variation of the means with a 95% confidence interval. With *p* < 0.05 as a significant difference, using the same statistical software GraphPad prism 9.0.0.

## Figures and Tables

**Figure 1 toxins-14-00598-f001:**
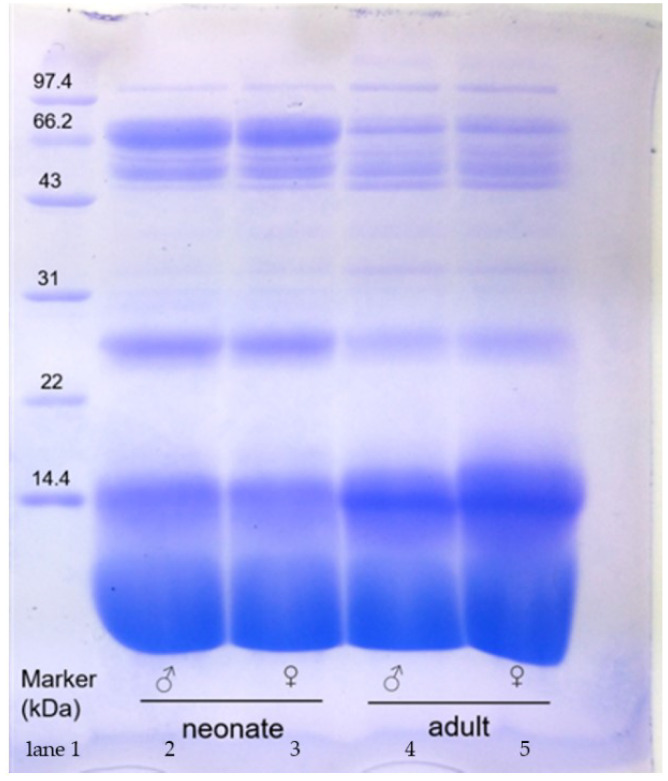
The 15% SDS-PAGE of *N. atra* crude venom (40 μg for each lane) under reducing. Lane 1: marker; Lane 2: neonate males; Lane 3: neonate females; Lane 4: adult males; Lane 5: adult females.

**Figure 2 toxins-14-00598-f002:**
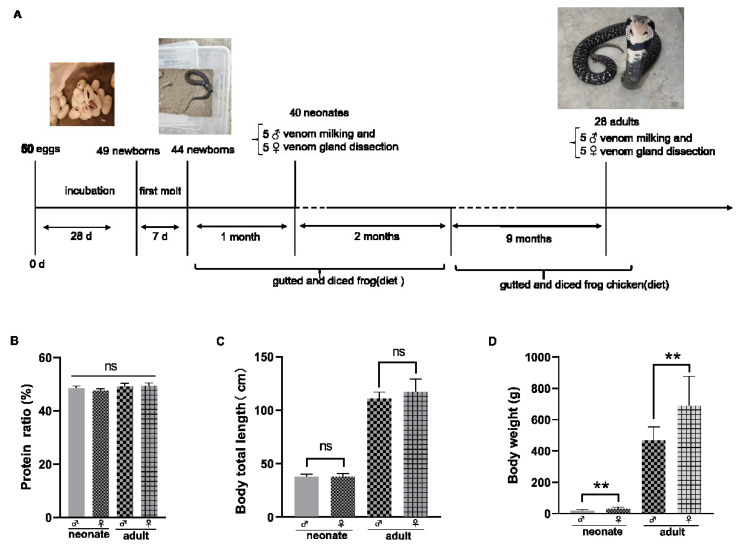
The breeding process of captive *N. atra* (**A**). Ratio of protein in crude venom (**B**). Snake body total length (**C**). Snake body weight (**D**). ♀: female; ♂: male.Significance analysis was performed by one-way ANOVA, *p* > 0.05 (ns), *p* < 0.002 (**).

**Figure 3 toxins-14-00598-f003:**
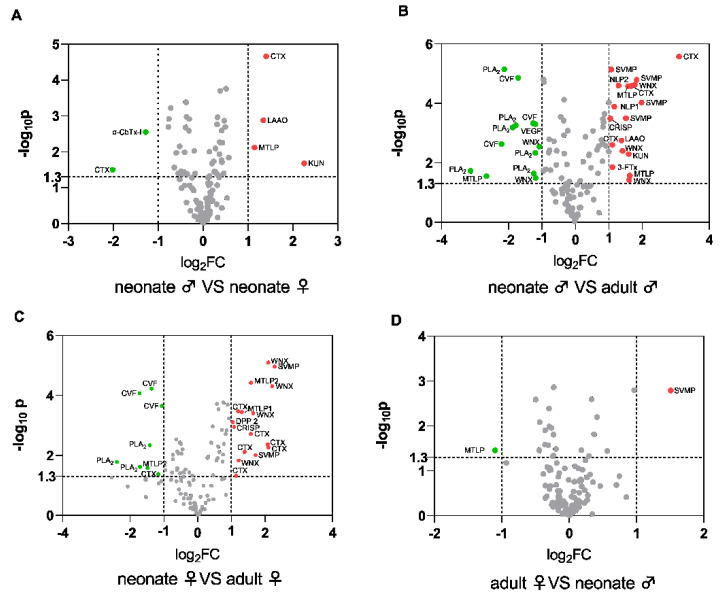
The relative differences of *N. atra* venomic groups. (**A**) Neonate males versus neonate females. (**B**) Neonate males versus adult males. (**C**) Neonate females versus adult females. (**D**) Adult females versus adult males. The red dots represent the upregulated and green dots present the downregulated proteins. *p* < 0.05(−log10 *p* > 1.3), |log2FoldChange| > 1. Abbreviations: 3-FTX, three-finger toxins; CRISP, cysteine-rich secretory protein; SVMP, snake venom metalloproteinase; CVF, cobra venom factor; PDE, phosphodiesterase; NGF, nerve growth factor; AChE, acetylcholinesterase; PLA_2_, phospholipase A_2_; NP, natriuretic peptide; VESP, vespryns; KUN, kunitz-type inhibitor; CTL, C-type lectin; 5′-NT, 5′-nucleotidase; QC, glutaminyl-peptide cyclotransferases; PLI, phospholipase A_2_ inhibitor; SVSP, snake venom serine protease; LAAO, L-amino acid oxidase; VEGF, vascular endothelial growth factor; DPP, dipeptidylpeptidase; HAase, hyaluronidase; PLB, phospholipase B; CTX, cytotoxin; WNX, weak neurotoxin; α-CbTx-I, type I alpha-neurotoxin (short neurotoxin, SNX); MTLP, Muscarinic toxin-like protein; NLP, neurotoxin like protein.

**Figure 4 toxins-14-00598-f004:**
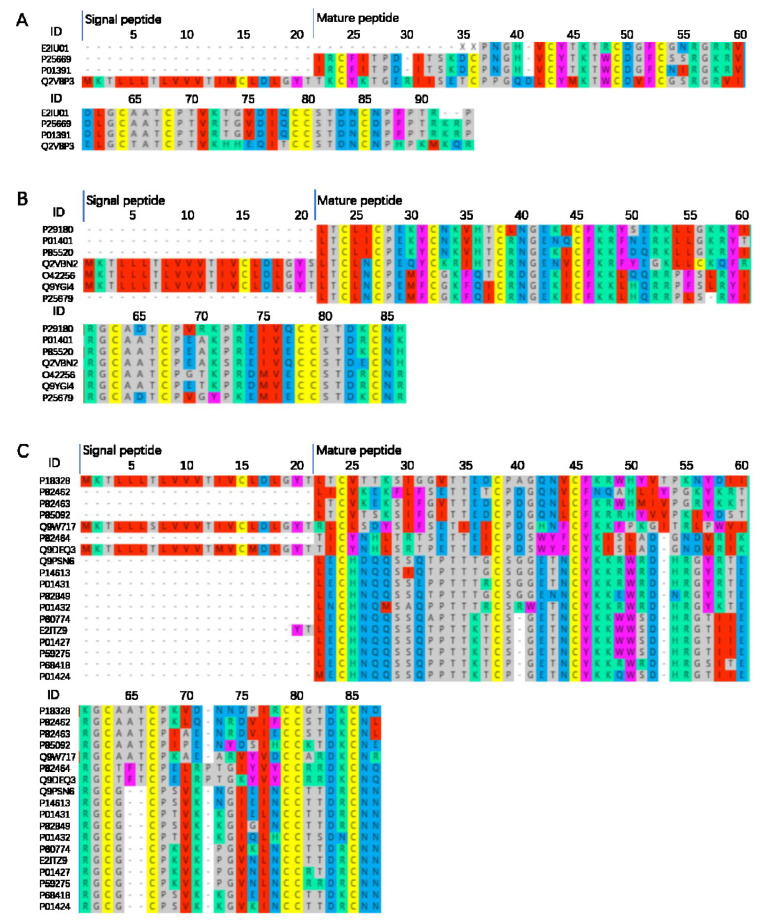
*N. atra* 3-FTx sequence alignment analysis. LNX (long neurotoxin) (**A**). WNX (weak neurotoxin) (**B**). SNX (short neurotoxin) (**C**). CTX (cytotoxin) (**D**). ID: the ID number of the protein in Uniprot. Online multi-sequence alignment by MAFFT Version 7, parameter default value.

**Figure 5 toxins-14-00598-f005:**
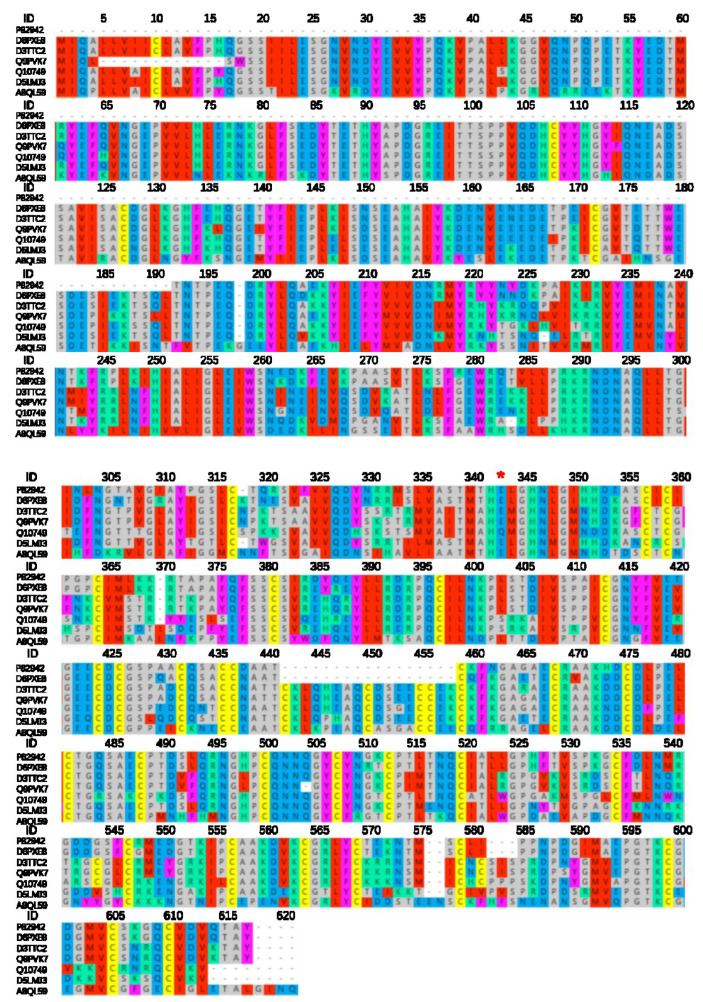
SVMP of *N. atra* sequence alignment analysis. This was compared online by MAFFT Version 7, parameter default value. ID: the ID number of the protein in Uniprot., *: active site.

**Figure 6 toxins-14-00598-f006:**
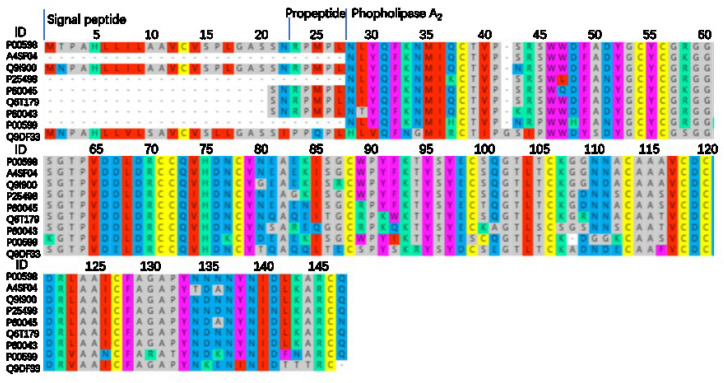
*N. atra* PLA_2_ sequence alignment analysis. This was compared online by MAFFT Version 7, parameter default value. ID: the ID number of the protein in Uniprot.

**Figure 7 toxins-14-00598-f007:**
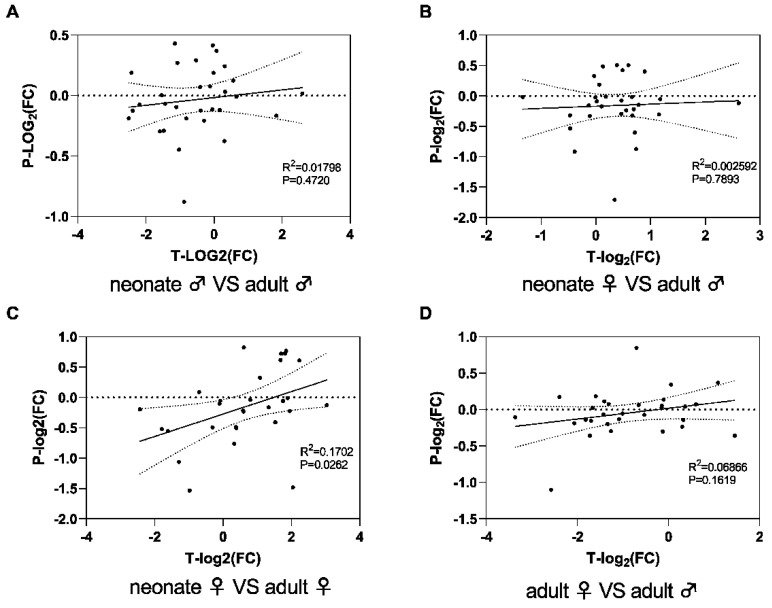
The correlation analysis of the proteome and venom gland transcriptome of *N. atra* venom. Neonate males versus adult males (**A**); neonate females versus neonate males (**B**); neonate females versus adult females (**C**); adult females versus adult males (**D**). T: transcript; P: protein; FC: fold change; R^2^: Pearson R squared.

**Figure 8 toxins-14-00598-f008:**
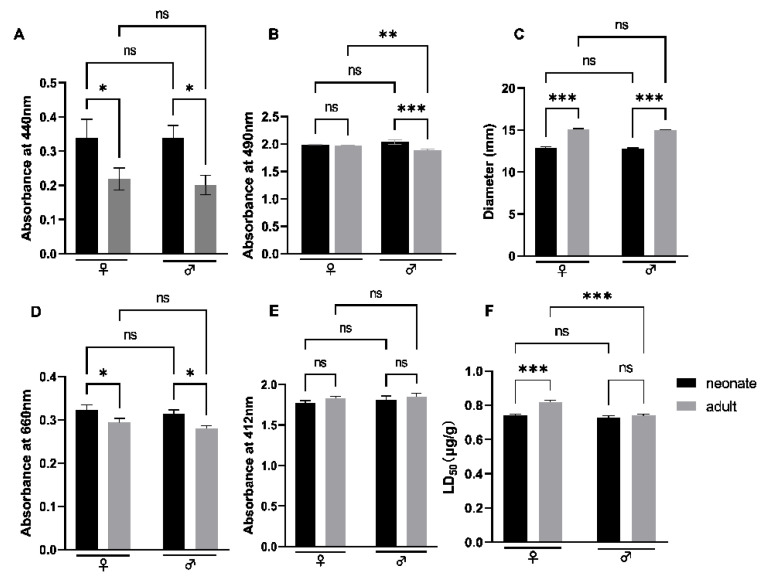
The enzyme activities of *N. atra* crude venom. SVMP (**A**); LAAO (**B**); PLA_2_ (**C**); 5′-NT (**D**); AChE (**E**); LD_50_ (**F**). ♀: female; ♂: male. Significance analyzed by Tukey’s multiple comparison test, the value of *p* > 0.05 (ns), *p* < 0.05 (*), *p* < 0.002 (**), *p* < 0.001 (***).

**Figure 9 toxins-14-00598-f009:**
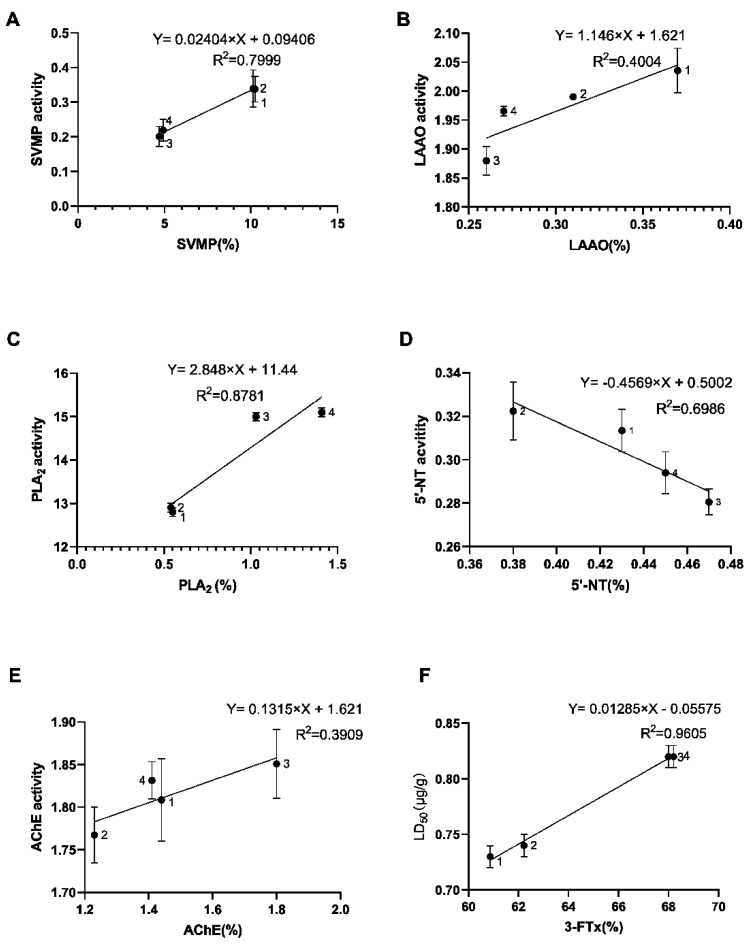
The correlation of *N. atra* venom protein ratios with corresponding enzyme activities. SVMP (**A**). LAAO (**B**). PLA_2_ (**C**). 5′-NT (**D**). AChE (**E**). LD_50_ (**F**). 1: neonate males; 2: neonate females; 3: adult males; 4: adult females. Significance analyzed by Simple linear regression analysis, the value of *p* < 0.05 considered significant.

**Table 1 toxins-14-00598-t001:** The list of proteins that were identified from *N. atra* snake venom by LC-MS/MS.

Protein Accessions	Description
V8N8G0	Alkaline phosphatase
ENSNNAP00000025215.1*	Acetylcholinesterase
Q9DF56	Acidic phospholipase A_2_
P00598	Acidic phospholipase A_2_ 1
Q9DF33	Acidic phospholipase A_2_ 2
ENSNNAP00000015782.1*	Acidic phospholipase A_2_ 2
P60045	Acidic phospholipase A_2_ 3
Q6T179	Acidic phospholipase A_2_ 4
Q9I900	Acidic phospholipase A_2_ D
P25498	Acidic phospholipase A_2_ E
A4FS04	Acidic phospholipase A_2_ natratoxin
P86542	Phospholipase A_2_ 3
P00599	Basic phospholipase A_2_ 1
P60043	Basic phospholipase A_2_ 1
V8ND68	Phospholipase B-like
Q7LZI1	Phospholipase A_2_ inhibitor 31 kDa subunit
ENSNNAP00000008460.1*	PLIalpha-like protein
V8NQ76	Atriopeptidase
V8N495	Carboxypeptidase
ENSNNAP00000003269.1*	Carboxypeptidase D
I2C090	Ophiophagus venom factor
Q91132	Cobra venom factor
Q91132	Cobra venom factor
ENSNNAP00000021869.1*	Cobra venom factor
ENSNNAP00000023282.1*	Cobra venom factor
ENSNNAP00000010931.1*	Complement C3
ENSNNAP00000013063.1*	Complement C3
Q01833	Complement C3
Q90WI6	C-type lectin BML-1
E3P6P4	Cystatin
P86543	Cysteine-rich venom protein
P84807	Cysteine-rich venom protein 25-A
P84808	Cysteine-rich venom protein kaouthin-2
Q7T1K6	Cysteine-rich venom protein natrin-1
Q53B46	Beta-cardiotoxin CTX15
P60305	Cytotoxin 1
P86541	Cytotoxin 10
Q9W6W6	Cytotoxin 10
ENSNNAP00000013731.1*	Cytotoxin 11
Q98956	Cytotoxin 1b
P85429	Cytotoxin 1f
P01442	Cytotoxin 2
Q9DGH9	Cytotoxin 2
P01445	Cytotoxin 2
P01463	Cytotoxin 2
O93472	Cytotoxin 2c
P01452	Cytotoxin 4
O93473	Cytotoxin 4a
O73856	Cytotoxin 4b
P07525	Cytotoxin 5
P01457	Cytotoxin 5
P24779	Cytotoxin 5
P25517	Cytotoxin 5
P24779	Cytotoxin 5
Q98965	Cytotoxin 6
P01465	Cytotoxin 6
P49122	Cytotoxin 7
O73859	Cytotoxin 7
P86540	Cytotoxin 8
P60311	Cytotoxin KJC3
P60308	Cytotoxin SP15c
P62377	Cytotoxin-like basic protein
P18328	Muscarinic toxin 2
P82462	Muscarinic toxin-like protein 1
P82463	Muscarinic toxin-like protein 2
P82464	Muscarinic toxin-like protein 3
Q9PSN6	Neurotoxin 3
Q9DEQ3	Neurotoxin homolog NL1
Q9W717	Neurotoxin-like protein NTL2
P60774	Short neurotoxin 1
P01427	Short neurotoxin 1
P14613	Short neurotoxin 1
P01431	Short neurotoxin 1
P68418	Short neurotoxin 1
P01424	Short neurotoxin 1
P01432	Short neurotoxin 3
E2IU01	Long neurotoxin 7
E2ITZ9	Cobrotoxin-b
P85092	Toxin AdTx1
P29180	Weak neurotoxin 6
O42256	Weak neurotoxin 6
Q2VBN2	Weak neurotoxin WNTX34
P01401	Weak toxin CM-11
P25679	Weak toxin CM-9a
P85520	Oxiana weak toxin
Q9YGI4	Probable weak neurotoxin NNAM2
P25669	Long neurotoxin 2
Q2VBP3	Long neurotoxin LNTX37
P01391	Alpha-cobratoxin
ENSNNAP00000012975.1*	Cobrotoxin
P82849	Cobrotoxin II
P59275	Cobrotoxin-b
D9IX97	Natriuretic peptide Na-NP
V8NF35	Dipeptidyl peptidase 2
V8P9G9	Venom dipeptidylpeptidase IV
ENSNNAP00000025792.1*	Dipeptidyl peptidase 7
V8P395	Glutathione peroxidase
ENSNNAP00000008637.1*	Hyaluronidase-2-like isoform X1
ENSNNAP00000024830.1*	Insulin like growth factor 1
ENSNNAP00000006918.1*	Insulin like growth factor binding protein 3
V8N7H9	BPTI/Kunitz domain-containing protein-like isoform X2
P20229	Kunitz-type serine protease inhibitor
ENSNNAP00000023266.1*	Kunitz-type serine protease inhibitor 4-like
A8QL51	L-amino acid oxidase
ENSNNAP00000002373.1*	L-amino-acid oxidase
A8QL58	L-amino-acid oxidase
V8P5W4	Putative serine carboxypeptidase CPVL
ENSNNAP00000011792.1*	Serpin family E member 2
A0A2I4HXH5	5′-nucleotidase
V8NYW9	5′-nucleotidase
A8QL53	Snake venom serine protease NaSP
P86545	Thrombin-like enzyme TLP
V8NCP7	Vascular endothelial growth factor C
P61899	Venom nerve growth factor
Q5YF90	Venom nerve growth factor 1
A0A2D0TC04	Venom phosphodiesterase
P83234	Vespryn-21
D3TTC2	Zinc metalloproteinase-disintegrin-like atragin
ENSNNAP00000015130.1*	Zinc metalloproteinase-disintegrin-like atragin
D5LMJ3	Zinc metalloproteinase-disintegrin-like atrase-A
ENSNNAP00000015250.1*	Zinc metalloproteinase-disintegrin-like atrase-A
ENSNNAP00000015377.1*	Zinc metalloproteinase-disintegrin-like atrase-A
D6PXE8	Zinc metalloproteinase-disintegrin-like atrase-B
Q9PVK7	Zinc metalloproteinase-disintegrin-like cobrin
A8QL59	Zinc metalloproteinase-disintegrin-like NaMP
Q10749	Snake venom metalloproteinase-disintegrin-like mocarhagin
P82942	Hemorrhagic metalloproteinase-disintegrin-like kaouthiagin

“*” Matched with *N. naja* snake veonom reference proteome (ftp://ftp.ensembl.org/pub/release-101/fasta/naja_naja/pep/, accessed on 22 January 2021) and the others matched with Uniprot (https://www.uniprot.org/). Accessed on 22 January 2021.

**Table 2 toxins-14-00598-t002:** The relative abundance of venom components in *N. atra* venom and quantification of the relative abundance of venom components using TMT label.

% Proteome	Neonate Males	Neonate Females	Adult Males	Adult Females
PLB	0.07%	0.06%	0.10%	0.10%
HAase	0.09%	0.10%	0.11%	0.11%
DPP-IV	0.14%	0.14%	0.18%	0.14%
VEGF	0.22%	0.22%	0.23%	0.21%
PLI	0.19%	0.28%	0.26%	0.30%
CTL	0.35%	0.37%	0.47%	0.51%
LAAO	0.38%	0.32%	0.24%	0.27%
5′-NT	0.43%	0.39%	0.40%	0.45%
QC	0.49%	0.50%	0.40%	0.39%
SVSP	0.50%	0.50%	0.26%	0.27%
Acidic PLA_2_	0.53%	0.52%	0.94%	1.3%
Basic PLA_2_	0.02%	0.02%	0.07%	0.1%
KUN	0.95%	0.70%	0.71%	0.73%
NP	1.05%	1.83%	1.03%	1.08%
VESP	1.12%	1.18%	0.72%	0.86%
AChE	1.44%	1.23%	1.80%	1.41%
NGF	2.52%	2.70%	2.82%	2.94%
PDE	3.49%	3.30%	4.35%	4.51%
CVF	3.53%	3.92%	4.70%	4.78%
SVMP(P-III)	10.23%	10.13%	5.02%	4.93%
CRISP	11.39%	9.37%	7.08%	6.65%
LNX(3-FTx)	0.10%	0.09%	0.13%	0.14%
SNX(3-FTx)	15.67%	15.73%	11.00%	13.65%
WNX(3-FTx)	5.08%	6.18%	5.09%	5.79%
CTX(3-FTx)	40.00%	40.22%	51.77%	48.63%

**Table 3 toxins-14-00598-t003:** Distribution of the SVMP glycosylation modification sites.

ID	Distribution of Glycosylation Sites (N-linked Asparagine)		
	Metalloprotease Structural Domains	Disintegrin Structural Domain	Cysteine-Rich Structural Domains
P82942	304	-	-
D6PXE8	320	-	509
D3TTC2	-	438	-
Q9PVK7	-	438	-
Q10749	303	-	509
D5LMJ3	221, 273, 304	438	527
A8QL59	225, 268, 319	-	551

- No data available. ID: ID number of the protein in Uniprot.

**Table 4 toxins-14-00598-t004:** The list of genes that were obtained from the identification of the *N. atra* venom gland transcriptome. All the genes co-existed in all the groups (FPKM > 1).

Gene ID	Gene Chromosome	Gene Coding Protein	Abbreviation
ENSNNAG00000009518	3	Alpha-elapitoxin-Nk2a	α-CbTx(3-FTx)
ENSNNAG00000009534	3	Muscarinic toxin-like protein 2	MTLP-2(3-FTx)
ENSNNAG00000011050	3	Muscarinic toxin-like protein 3 homolog	MTLP-3(3-FTx)
ENSNNAG00000009404	3	Cytotoxin 3a	CTX(3-FTx)
ENSNNAG00000009217	3	Cytotoxin 4N	CTX(3-FTx)
ENSNNAG00000011032	3	Cytotoxin 2	CTX(3-FTx)
ENSNNAG00000011010	3	Cytotoxin 5	CTX(3-FTx)
ENSNNAG00000008699	3	Probable weak neurotoxin NNAM1	WNX(3-FTx)
ENSNNAG00000008719	3	Tryptophan-containing weak neurotoxin	WNX(3-FTx)
ENSNNAG00000009048	3	Neurotoxin-like protein NTL2	NTL2(3-FTx)
ENSNNAG00000011613	SOZL01001066.1	Neurotoxin homolog NL1	NL1(3-FTx)
ENSNNAG00000011709	SOZL01001066.1	Cardiotoxin 7	CTX(3-FTx)
ENSNNAG00000008731	3	Cobrotoxin	CBT(3-FTx)
ENSNNAG00000007474	1	Snake venom 5′-nucleotidase	5′ NT
ENSNNAG00000007520	1	Snake venom 5′-nucleotidase	5′ NT
novel.2091	SOZL01000663.1	5′-nucleotidase	5′ NT
ENSNNAG00000018750	2	5′-nucleotidase	5′ NT
ENSNNAG00000016482	SOZL01001525.1	Acetylcholinesterase	AchE
ENSNNAG00000016482	SOZL01001525.1	Acetylcholinesterase	AchE
ENSNNAG00000015104	MIC_6	Aminopeptidase	AP
ENSNNAG00000008828	2	Complement C3	C3
ENSNNAG00000008657	2	Complement C3	C3
ENSNNAG00000012844	SOZL01001814.1	Cathelicidin-related peptide	CATH
ENSNNAG00000014282	1	Cysteine-rich venom protein ophanin	CRISP
ENSNNAG00000014151	1	Cysteine-rich venom protein ophanin	CRISP
ENSNNAG00000018595	MIC_4	Cysteine-rich secretory protein	CRISP
ENSNNAG00000000526	2	C-type lectin	CTL
ENSNNAG00000018045	7	C-type lectin lectoxin-Thr1	CTL
ENSNNAG00000000430	2	C-type lectin	CTL
ENSNNAG00000009410	1	C-type lectin	CTL
ENSNNAG00000018053	7	C-type lectin	CTL
ENSNNAG00000016651	MIC_7	C-type lectin	CTL
novel.1968	MIC_9	Cystatin (cysteine proteinase inhibitor)	CYS
ENSNNAG00000010061	2	Cystatin-B	CYS
ENSNNAG00000005504	MIC_9	Cystatin	CYS
ENSNNAG00000012223	3	Cystatin-2	CYS
novel.1967	MIC_9	Cystatin	CYS
ENSNNAG00000006541	1	Dipeptidyl peptidase IV	DPP-IV
ENSNNAG00000013472	2	Glutathione peroxidase 3	GPX
ENSNNAG00000013545	1	Glutathione peroxidase 2	GPX
ENSNNAG00000013509	2	Glutathione peroxidase 6	GPX
ENSNNAG00000008046	2	Glutathione peroxidase 1	GPX
ENSNNAG00000002583	2	Hyaluronidase	HAase
ENSNNAG00000005815	MIC_2	Hyaluronidase	HAase
ENSNNAG00000018577	MIC_3	Kunitz-type serine protease inhibitor	KSPI
ENSNNAG00000018577	MIC_3	Kunitz-type protease inhibitor	KSPI
ENSNNAG00000015199	3	Kunitz-type protease inhibitor	KSPI
ENSNNAG00000001233	1	Kunitz-type protease inhibitor	KSPI
ENSNNAG00000001705	2	L-amino-acid oxidase	LAAO
ENSNNAG00000001808	2	L-amino-acid oxidase	LAAO
ENSNNAG00000001642	2	L-amino-acid oxidase	LAAO
ENSNNAG00000001705	2	L-amino-acid oxidase	LAAO
ENSNNAG00000013519	MIC_2	Nerve growth factor	NGF
ENSNNAG00000004808	3	Venom nerve growth factor	NGF
ENSNNAG00000004800	3	Venom nerve growth factor1	NGF
ENSNNAG00000004808	3	Venom nerve growth factor	NGF
ENSNNAG00000006727	4	Natriuretic peptide Na-NP	NP
ENSNNAG00000006779	4	Natriuretic peptide Na-NP	NP
ENSNNAG00000006809	4	Natriuretic peptide Na-NP	NP
ENSNNAG00000006727	4	Natriuretic peptide Na-NP	NP
ENSNNAG00000000097	1	Venom phosphodiesterase 1	PDE
ENSNNAG00000013762	MIC_11	Group IIE secretory phospholipase A_2_	PLA_2_
ENSNNAG00000003407	1	Phospholipase A_2_ crotoxin basic subunit	PLA_2_
ENSNNAG00000010611	SOZL01000342.1	Acidic phospholipase A_2_	PLA_2_
ENSNNAG00000003618	4	Phospholipase A_2_	PLA_2_
ENSNNAG00000015696	7	Phospholipase B	PLB
novel.1890	MIC_8	Phospholipase B	PLB
novel.1889	MIC_8	Phospholipase B	PLB
ENSNNAG00000005694	4	PLIalpha-like protein	PLI
ENSNNAG00000003592	MIC_3	Phospholipase A_2_ inhibitor	PLI
ENSNNAG00000003672	MIC_3	Phospholipase A_2_ inhibitor	PLI
ENSNNAG00000013551	2	Snake venom serine protease NaSP	SVSP
ENSNNAG00000015301	MIC_3	Snake venom serine protease	SVSP
ENSNNAG00000015396	MIC_3	Snake venom serine protease	SVSP
ENSNNAG00000017712	6	Venom prothrombin activator	VPA
ENSNNAG00000017242	1	Vascular endothelial growth factor A	VEGF
ENSNNAG00000011309	MIC_5	Vascular endothelial growth factor	VEGF
ENSNNAG00000008620	MIC_9	Vascular endothelial growth factor A	VEGF
ENSNNAG00000004103	MIC_2	Cysteine-rich with EGF-like domain	CREGF
ENSNNAG00000001343	2	Cysteine-rich with EGF-like domain protein	CREGF
ENSNNAG00000007968	4	Vascular endothelial growth factor C	VEGF
ENSNNAG00000017242	1	Vascular endothelial growth factor A	VEGF
ENSNNAG00000013375	SOZL01001403.1	Cobra venom factor	CVF
ENSNNAG00000008178	2	Cobra venom factor	CVF
ENSNNAG00000013375	SOZL01001403.1	Cobra venom factor	CVF
ENSNNAG00000008178	2	Cobra venom factor	CVF
novel.1103	3	Snake toxin and toxin-like protein	STLK
novel.1007	3	Snake toxin and toxin-like protein	STLK
novel.1104	3	Snake toxin and toxin-like protein	STLK
novel.2094	SOZL01000688.1	Snake toxin and toxin-like protein	STLK
ENSNNAG00000009894	MIC_1	Zinc metalloproteinase-disintegrin-like NaMP	SVMP
ENSNNAG00000017863	MIC_8	Disintegrin and metalloproteinase domain-containing protein 9	SVMP
ENSNNAG00000013917	2	A disintegrin and metalloproteinase with thrombospondin motifs 12	SVMP
ENSNNAG00000011023	Z	Disintegrin and metalloproteinase domain-containing protein 11	SVMP
ENSNNAG00000010003	MIC_1	Zinc metalloproteinase-disintegrin-like atragin	SVMP
ENSNNAG00000008597	MIC_6	A disintegrin and metalloproteinase with thrombospondin motifs 17	SVMP
ENSNNAG00000005308	MIC_1	Disintegrin and metalloproteinase domain-containing protein 9	SVMP
novel.1209	4	Zinc metalloprotease	SVMP

**Table 5 toxins-14-00598-t005:** The relative abundance of venom components of *N. atra* gland venom transcriptomes. Quantification of relative abundance of venom components by TMT label.

FPKM%	Neonate Males	Neonate Females	Adult Males	Adult Females
PLI	0.001%	0.001%	0.001%	0.004%
C3	0.005%	0.002%	0.007%	0.008%
CREGF	0.015%	0.012%	0.006%	0.053%
AP	0.015%	0.013%	0.007%	0.062%
DPP-IV	0.017%	0.007%	0.014%	0.022%
VEGF	0.027%	0.015%	0.013%	0.059%
PLB	0.039%	0.037%	0.053%	0.083%
HAase	0.057%	0.031%	0.067%	0.067%
VPA	0.069%	0.025%	0.010%	0.063%
CATH	0.072%	0.006%	0.011%	0.022%
AChE	0.091%	0.072%	0.049%	0.167%
CYS	0.107%	0.081%	0.064%	0.172%
PDE	0.112%	0.059%	0.134%	0.136%
SVSP	0.168%	0.124%	0.145%	0.210%
LAAO	0.213%	0.121%	0.055%	0.207%
KUN	0.221%	0.151%	0.203%	0.221%
CVF	0.223%	0.096%	0.213%	0.179%
CTL	0.253%	0.212%	0.177%	0.456%
5′-NT	0.347%	0.211%	0.154%	0.400%
CRISP	0.615%	0.346%	0.135%	0.497%
NGF	0.972%	0.887%	1.386%	1.450%
SVMP	0.985%	0.515%	0.078%	0.653%
Basic PLA_2_	1.305%	1.244%	3.405%	1.829%
Acidic PLA_2_	0.001%	0.001%	0.001%	0.001%
STLK	1.673%	2.182%	1.853%	3.056%
GPX	2.352%	1.521%	1.386%	2.593%
NP	4.738%	3.806%	2.556%	6.932%
LNX(3-FTx)	0.751%	2.630%	1.918%	7.968%
CTX(3-FTx)	37.825%	38.359%	26.955%	29.696%
WNX(3-FTx)	12.815%	9.078%	3.417%	3.429%
SNX(3-FTx)	33.919%	38.158%	55.530%	39.307%

Abbreviations: 3-FTx, three-finger toxins; NP, natriuretic peptide; STLK, snake toxin and toxin-like protein; GPX, glutathione peroxidase; PLA_2_, phospholipase A_2_; NGF, nerve growth factor; SVMP, snake venom metalloproteinase; CRISP, cysteine-rich secretory protein; CTL, C-type lectin; 5′-NT, 5′-nucleotidase; KUN, kunitz-type inhibitor; SVSP, snake venom serine protease; LAAO, L-amino acid oxidase; CVF, cobra venom factor; CYS, cystatin; AChE, acetylcholinesterase; PDE, phosphodiesterase; PLB, phospholipase B; HAase, hyaluronidase; VPA, venom prothrombin activator; AP, aminopeptidase; CREGF, cysteine-rich with EGF-like domain protein; VEGF, vascular endothelial growth factor; DPP IV, dipeptidylpeptidase IV; CATH, cathelicidin-related peptide; C3, complement C3; PLI, phospholipase A_2_ inhibitor; CTX, cytotoxin; WNX, weak toxin; SNX, short neurotoxin; LNX, long neurotoxin; MTLP, muscarinic toxin-like protein.

**Table 6 toxins-14-00598-t006:** Matched *N. atra* snake venom proteome and venom gland transcriptome.

Protein Accessions	Gene	Protein Name
A0A2D0TC04	ENSNNAG00000000097	Venom phosphodiesterase
A8QL53	ENSNNAG 00000013551	Snake venom serine protease NaSP
	ENSNNAG00000015396	Snake venom serine protease NaSP
A8QL58	ENSNNAG00000001808	L-amino-acid oxidase
A8QL59	ENSNNAG00000009894	Zinc metalloproteinase-disintegrin -like NaMP
	ENSNNAG00000008597	Zinc metalloproteinase-disintegrin -like NaMP
D5LMJ3	ENSNNAG00000005308	Zinc metalloproteinase-disintegrin -like atrase-A
	ENSNNAG00000013917	Zinc metalloproteinase-disintegrin -like atrase-A
D9IX97	ENSNNAG00000006809	Natriuretic peptide Na-NP
	ENSNNAG00000006727	Natriuretic peptide Na-NP
	ENSNNAG00000006779	Natriuretic peptide Na-NP
E3P6P4	ENSNNAG00000005504	Cystatin
ENSNNAP00000002373.1	ENSNNAG00000001642	L-amino-acid oxidase (Fragment chain B)
	ENSNNAG00000001705	L-amino-acid oxidase
ENSNNAP00000008460.1	ENSNNAG00000005694	PLI alpha-like protein
ENSNNAP00000008637.1	ENSNNAG00000005815	Hyaluronidase-2-like isoform X1
	ENSNNAG00000002583	Hyaluronidase-2-like isoform X1
ENSNNAP00000009791.1	ENSNNAG00000006541	Venom dipeptidylpeptidase IV
ENSNNAP00000011851.1	ENSNNAG00000008620	Vascular endothelial growth factor C
	ENSNNAG00000007968	Vascular endothelial growth factor C
	ENSNNAG00000017242	Vascular endothelial growth factor C
ENSNNAP00000012520.1	ENSNNAG00000008178	Cobra venom factor
ENSNNAP00000012975.1	ENSNNAG00000008731	Cobrotoxin
ENSNNAP00000013063.1	ENSNNAG00000008657	Complement C3
ENSNNAP00000014120.1	ENSNNAG00000009518	Alpha-cobratoxin
ENSNNAP00000015130.1	ENSNNAG00000010003	Zinc metalloproteinase-disintegrin -like atragin
ENSNNAP00000021869.1	ENSNNAG00000013375	Cobra venom factor
ENSNNAP00000023044.1	ENSNNAG00000001233	BPTI/Kunitz domain-containing protein-like isoform X2
	ENSNNAG00000015199	BPTI/Kunitz domain-containing protein-like isoform X2
ENSNNAP00000025215.1	ENSNNAG00000016482	Acetylcholinesterase
P00598	ENSNNAG00000010611	Acidic phospholipase A2 1
P01442	ENSNNAG00000009217	Cytotoxin 2
	ENSNNAG00000011010	Cytotoxin 2
P01445	ENSNNAG00000011032	Cytotoxin 2
P29180	ENSNNAG00000008699	Weak neurotoxin 6
P61899	ENSNNAG00000004808	Venom nerve growth factor
P62377	ENSNNAG00000011709	Cytotoxin-like basic protein
P86540	ENSNNAG00000009404	Cytotoxin 8
Q01833	ENSNNAG00000008828	Complement C3
Q10749	ENSNNAG00000017863	Snake venom metalloproteinase -disintegrin-like mocarhagin
	ENSNNAG00000011023	Snake venom metalloproteinase -disintegrin-like mocarhagin
Q5YF90	ENSNNAG00000004800	Venom nerve growth factor 1
	ENSNNAG00000013519	Venom nerve growth factor 1
Q6T179	ENSNNAG00000013762	Acidic phospholipase A2 4
	ENSNNAG00000003407	Acidic phospholipase A2 4
Q7LZI1	ENSNNAG00000003592	Phospholipase A2 inhibitor 31 kDa subunit
Q7T1K6	ENSNNAG00000014151	Cysteine-rich venom protein natrin-1
Q9DEQ3	ENSNNAG00000011613	Neurotoxin homolog NL1
	ENSNNAG00000011050	Neurotoxin homolog NL1
Q9W717	ENSNNAG00000009048	Neurotoxin-like protein NTL2
Q9YGI4	ENSNNAG00000008719	Probable weak neurotoxin NNAM2
V8P395	ENSNNAG00000013472	Glutathione peroxidase
	ENSNNAG00000008046	Glutathione peroxidase
	ENSNNAG00000013545	Glutathione peroxidase

## Data Availability

The raw data that were used in this study and the statistical analyses are available via the corresponding author at Chongqing Normal University.

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
