# Peer review of "Venom Variation of Neonate and Adult Chinese Cobras in Captivity Concerning Their Foraging Strategies"

_toxins, 2022, doi:10.3390/toxins14090598_

Round 1

Reviewer 1 Report

Review Report Toxins

Title: Venom variation of neonate and adult Chinese cobras in captive concerning their foraging strategies”

Brief summary

The manuscript focuses on proteomic (with LC-MS/MS)  and trascriptomic profile determination (with Illumina platform sequencing)  of Chinese cobras (N. atra) venom during ontogenetic changes and concerning their foraging strategies, by emphasizing differences between female and male as well as between adults and neonates in wild and captive specimens. Authors supported that currently there is no report “omics” study on captive Chinese cobra and they concluded that levels of different types of toxin families were deeply altered in wild and captive specimens, and, in particular, 10 new transcripts and 88 unique transcripts were identified.

Broad comments

The manuscript focuses on proteomic (with LC-MS/MS) and trascriptomic profile determination (with Illumina platform sequencing) of Chinese cobras (N. atra) venom during ontogenetic changes and concerning their foraging strategies, by emphasizing differences between female and male as well as between adults and neonates in wild and captive specimens. Authors supported that currently there is no report “omics” study on captive Chinese cobra and they concluded that levels of different types of toxin families were deeply altered in wild and captive specimens, and, in particular, 10 new transcripts and 88 unique transcripts were identified. The experimental design of the study is robust, but along the text there are a lot grammatical and typing mistakes. Furthermore the number of animals enrolled especially for proteomic characterization and transcriptome sequencing is not clearly evident from materials and methods section.

Introduction

My suggestion is to implement introduction in particular with information about the risks associated to snakebites in worldwide population and with proteomic/transcriptomic studies conducted until now on Cobra snake venom, with updated references.

Results

2.3 Protein sequence alignment

This section and is paragraph is too long and too detailed, probably not proportionate with the rest of the text. Furthermore Figure 3 is too far from the text.

Discussion

Lines 352-360: this paragraph include the aim and the framing of the topic, so please move from this section

Lines 485-489: this passage is not clear.

As reported in broad comments, along the text there are a lot grammatical and typing mistakes, in particular errors in punctuation and missing verbs with fragmented and incomplete sentences. Following some examples: line 69; line 103; line 152; line 156; line 219; line 339; line 340; line 375; line 397; line 408; lines 472-473; line 510; line 514; line 607; line 647.... There are also errors in plural terms (adult instead of adults) and in Table S1 where peiptide insetad of petide is reported.

English revision is recommended.

References

Please delete the journal reference guidelines from the text and check well the reported references in accordance with the format required by “Toxins-MDPI” Journal.

Author Response

Dear Reviewer
    Thank you for taking the time and effort to review my paper.

Best regard

Reviewer 2 Report

The MS entitled” Venom variation of neonate and adult Chinese cobras in captive concerning their foraging strategies” explores the differences at physiological, transcriptomic and proteomic level between neonate and adults specimens in captivity with those in the wild. They found important differences, being  3FTx  the most abundant components in adults whereas SVMP and CRISP are the most abundant in neonates. The findings of this work present clear evidence that antivenoms must include as immunogens venoms from both neonates and adults.

Authors clearly articulate the objectives of the work with clear testable hypothesis: the study is well designed. Conclusions are supported by the data. However, the manuscript is presented in an unintelligible fashion and therefore difficult to follow.

My only recommendation for authors is to hire professional translation services with editing, this would make the MS acceptable for review.

Author Response

The MS entitled” Venom variation of neonate and adult Chinese cobras in captive concerning their foraging strategies” explores the differences at physiological, transcriptomic and proteomic level between neonate and adults specimens in captivity with those in the wild. They found important differences, being  3FTx  the most abundant components in adults whereas SVMP and CRISP are the most abundant in neonates. The findings of this work present clear evidence that antivenoms must include as immunogens venoms from both neonates and adults.

Authors clearly articulate the objectives of the work with clear testable hypothesis: the study is well designed. Conclusions are supported by the data. However, the manuscript is presented in an unintelligible fashion and therefore difficult to follow.

My only recommendation for authors is to hire professional translation services with editing, this would make the MS acceptable for review.

Answer:I checked and revised as your suggestion.

Reviewer 3 Report

The authors examined the toxin profiles of infants and adults in artificial captivity for cobras inhabiting China and showed different profiles from wild species. This is a very interesting research treatise. As a reviewer, I think that the reader's understanding will be increased by making the following minor corrections.

1. Describe the limitation item in this paper in the discussion.

2. The "E" in (Figure 3E) of L134 is not in Figure 3, so correct it.

3. Table 6 of L312 is considered to refer to Table 8 in the text, so correct it.

4. Add a description of "F" to the description of Figure 8 of L344.

Author Response

The authors examined the toxin profiles of infants and adults in artificial captivity for cobras inhabiting China and showed different profiles from wild species. This is a very interesting research treatise. As a reviewer, I think that the reader's understanding will be increased by making the following minor corrections.

  1. Describe the limitation item in this paper in the discussion.
    Answer:I added limitation item in this paper in the discussion.
  2. The "E" in (Figure 3E) of L134 is not in Figure 3, so correct it.

Answer:I revised as your suggestion.

  1. Table 6 of L312 is considered to refer to Table 8 in the text, so correct it.

Answer:I revised as your suggestion.

  1. Add a description of "F" to the description of Figure 8 of L344.

Answer:I revised as your suggestion.

Reviewer 4 Report

Comments are inserted in pdf file.

Author Response

Dear Reviewer
    Thank you for taking the time and effort to review my paper.
    The four areas you mentioned in your review comments that need additional instrumentation information have been added in full.

Best regard

Round 2

Reviewer 2 Report

This is a great work that unveils interesting data. I have some particular comments on this version, however.

Lines 5-7

Please re-phrase. Could be something like: Here, LC-MS/MS and illumina technology were used to unveil the venome and trascriptome of neonates and adults N. atra specimens.  

Lines 8.

Consequently, is not the appropriate adverb. 

Lines 9.

What about: “The most abundant components of the venom were three finger toxins (3-FTx)…

Line 9

Typo: coun-tries

Line 32

Correct order based on number and order mentioned later on the text: Most of these bites occur in Asia, Africa and Latin America….

 Line 39

Please use a more appropriate reference. There are a few that point out the worldwide epidemiology of snakebite.  

Lines 43-46

The sentence must be rephrased since is confusing.

Lines 48

The following sentence lack of a proper adverb. This to make the whole paragraph more cohesive: “Plasticity of snake venom pose a huge challenge to the treatment of snakebite"

Line 49

Snake antivenom is the ONLY treatment proved affective to treat snakebite envenoming.

Line 49.

Antivenoms are made of antibodies harvested from the serum of hyperimmune animals, typically horses…

In the phrase:

“Antivenom is not always effective against the various complications associated with snakebites [19]. In addition, there are cases where not all venom components are targeted by antivenom, especially the low molecular mass protein component, three-finger toxin (3- 55 FTx), which is a major contributor to death in patients with cobra snake envenomation [20].” Please indicate what are the complications ? and cite works that expose 3ftx as main targets for Elapid antivenoms, particularly.  

Lines 57-60

Please use the corresponding cite.

 Please re-phrase: 

“Meanwhile, the standardized antivenom that fail to neutralize some venom components and responsible for the increase of victim mortality and anti-venom application dose [21]”

Lines 63-65

“The mortality rate in patients with snakebite in saw-scaled vipers has increased from < 2% to 10%-12% due to changes in snake venom fractions that have led to a decrease in the efficacy of antivenom”

This phrase is out of the context of Asian snakes. Furthermore, snake venom fractions do not vary, what could change in the antivenom manufacturing is the pool of venoms that were used. Please discuss, amend, or omit.

Lines 65-71

This number of phrases are not precisely well developed. There are plenty of new works that assessed animal-derived polyvalent antivenoms, that uses recombinant toxins and recombinant antibodies including synthetic antibodies., I recommend  to read and to include the data and to cite.  

Lines 70-71

Therefore, the identification of snake venom profile A hinge to treat snakebite effectively.

Lines 73-75

This traditional strategy falls into VENOMICS studies (term coined by many authors doing venom fractionation, MS, etc.). Therefore, the next phrase is not accurate. Please explain otherwise amend. 

Lines 87-90

Authors mentioned first 1B . I think this should be named 1A and make the corresponding changes in the figure. 
What is the reason of Fig 1A ? I couldn’t find any discussion all over the text..

Lines 92-93

Is not more appropriated to say age instead of growth period?

Lines 101-105

Please double check the syntaxis. 

Line 106

What if authors start your phrase using: “our results show that N.atra venom mainly consists …..

Lines 113-125

Please rephrase and make it more cohesive. There is no need to use the words “specimen” and “compared” in every sentence. 

Line 132 

This is redundant: “The relative differences between different..”

In the same Figure: is there a way to identify in the graphs each gender?  

Line 143  

There is a capital letter in the wrong place. :”3-FTx are the main components that make up the snake venom proteins and Responsible for snakebite lethality”

Please cite.

Line 144 -152

Firstly, it contains many grammatical errors, and it can be also shortened to make it clear and legible. Secondly, the figure list must follow an order. I.e. first goes 3A, 3B and so on. Please amend this in the text and modify the figure. Finally, please cite.  

Lines 158 -166

Although this is important, there is no need to discuss the mechanism of each toxin, rather, I would use the corresponding section to do it.

Lines 172 -173

Edition error.

Lines 201-204.

This sentence can be improved. 

First goes table 3 and then Figure 4, as authors indicate and discuss in the text.

Authors Unlike the other toxins families, authors did not provide SVMP sequence alignment nor mentioned the mechanism of action. Any reason?

Table 4 and 5.

In my opinion a graph like pie, donut or even dot plot can be used to visualize your data. It’ll improve the understanding and content of the tables, which the latter could be include as a supplementary table.

Line 296.

In my opinion the Section 2.6 should be the first section. This due to the level of detail and data that you obtain – following the rule from general to in-detail analysis.

Line 303 -306.

Densimetric more strongly, what does it mean? Couple of bands means 2 not 4…

Line 316-318

Why authors start explaining fig 8 F instead 8A?? please be consistent and follow a logical order. 

Figure 8 does not mentioned F

Line 320-323

Please check the grammar and syntaxis. 

Line 335

Citation missed in the sentence: “Intraspecific and interspecific variation in snake venom is common” 

Please re-phrase:

“In this study, there were significant differences between venom types of neonate and adult snakes, among which the venom content of neonate snakes was significantly higher than that of  adult ones SVMP and CRISP; 3-FTX, CVF, and PLA2 were significantly higher in adult snake than in neonate snake. These results were generally consistent with changes in the venom composition of wild Chinese cobra neonate specimens and adult specimens, also from Guangxi Province [8]”  The whole part is confusing and could be misleading. Please rephrase it , and edit it to make it simple and straight. 

Line 341 344

“Confirmed the existence of differences….” Sounds better: Confirmed the differences…

Please rephrase edit he whole sentence.

Overall, the discussion is aligned with the findings and authors make clear the limitations. The MS needs more work on the grammar since the number of typos and syntactic errors are common throughout the Text. Finally, edition is also needed to make, particularly in the result and discussion section to make it more fluid and therefore cohesive. It will be great to see this MS published soon. 

Author Response

Dear Reviewer
Thank you for your guidance. The author's note is attached and pls check.

Best regard
